# Age acquired skewed X chromosome inactivation is associated with adverse health outcomes in humans

Amy L Roberts[1]*, Alessandro Morea[1,2†], Ariella Amar[2], Antonino Zito[1‡, §], Julia S El-Sayed Moustafa[1], Max Tomlinson[1,2], Ruth CE Bowyer[1], Xinyuan Zhang[1], Colette Christiansen[1], Ricardo Costeira[1], Claire J Steves[1], Massimo Mangino[1,3], Jordana T Bell[1], Chloe CY Wong[4], Timothy J Vyse[2], Kerrin S Small[1]*

[1]Department of Twin Research & Genetic Epidemiology, King's College London, London, United Kingdom; [2]Department of Medical and Molecular Genetics, King's College London, London, United Kingdom; [3]NIHR Biomedical Research Centre, Guy's and St Thomas' Foundation Trust, London, United Kingdom; [4]Social, Genetic & Developmental Psychiatry Centre, Institute of Psychiatry, Psychology & Neuroscience, King's College London, London, United Kingdom

**\*For correspondence:** amy.roberts@kcl.ac.uk (ALR); kerrin.small@kcl.ac.uk (KSS)

**Present address:** †The FIRC Institute of Molecular Oncology, Italian Foundation for Cancer Research, Milan, Italy; ‡Department of Molecular Biology, Massachusetts General Hospital, Boston, United States; §Department of Genetics, Harvard Medical School, Boston, United States

## Abstract

**Background:** Ageing is a heterogenous process characterised by cellular and molecular hallmarks, including changes to haematopoietic stem cells and is a primary risk factor for chronic diseases. X chromosome inactivation (XCI) randomly transcriptionally silences either the maternal or paternal X in each cell of 46, XX females to balance the gene expression with 46, XY males. Age acquired XCI-skew describes the preferential selection of cells across a tissue resulting in an imbalance of XCI, which is particularly prevalent in blood tissues of ageing females, and yet its clinical consequences are unknown.

**Methods:** We assayed XCI in 1575 females from the TwinsUK population cohort using DNA extracted from whole blood. We employed prospective, cross-sectional, and intra-twin study designs to characterise the relationship of XCI-skew with molecular and cellular measures of ageing, cardiovascular disease risk, and cancer diagnosis.

**Results:** We demonstrate that XCI-skew is independent of traditional markers of biological ageing and is associated with a haematopoietic bias towards the myeloid lineage. Using an atherosclerotic cardiovascular disease risk score, which captures traditional risk factors, XCI-skew is associated with an increased cardiovascular disease risk both cross-sectionally and within XCI-skew discordant twin pairs. In a prospective 10 year follow-up study, XCI-skew is predictive of future cancer incidence.

**Conclusions:** Our study demonstrates that age acquired XCI-skew captures changes to the haematopoietic stem cell population and has clinical potential as a unique biomarker of chronic disease risk.

**Funding:** KSS acknowledges funding from the Medical Research Council [MR/M004422/1 and MR/R023131/1]. JTB acknowledges funding from the ESRC [ES/N000404/1]. MM acknowledges funding from the National Institute for Health Research (NIHR)-funded BioResource, Clinical Research Facility and Biomedical Research Centre based at Guy's and St Thomas' NHS Foundation Trust in partnership with King's College London. TwinsUK is funded by the Wellcome Trust, Medical Research Council, European Union, Chronic Disease Research Foundation (CDRF), Zoe Global Ltd and the National Institute for Health Research (NIHR)-funded BioResource, Clinical Research Facility and Biomedical Research Centre based at Guy's and St Thomas' NHS Foundation Trust in partnership with King's College London.

### Editor's evaluation

XCI skewing is affected by age but how this may affect a person's health is not known. Roberts et al., demonstrate that these changes result in an increased risk of cardiovascular disease and cancer. These findings will be of interest to researchers studying the impact of age on health.

## Introduction

Ageing is a heterogenous process characterised by cellular and molecular hallmarks and can manifest clinically as frailty and multimorbidity (*Clegg et al., 2013*; *López-Otín et al., 2013*). Ageing is a primary risk factor for diseases such as cardiovascular disease and cancer, and a better understanding of the biomarkers of ageing promises to reduce the burden of chronic disease which significantly impacts the human healthspan (*López-Otín et al., 2013*).

X chromosome inactivation (XCI) evolved in placental mammals to compensate for the X-linked gene dosage between XX females and XY males. XCI transcriptionally silences either the maternal or paternal X in each cell to equalise the gene expression between 46, XX females and 46, XY males (*Lyon, 1961*). The selection of which X is silenced is a random process that occurs during development, with the XCI status then clonally inherited by all daughter cells. Therefore, mammalian female tissues are mosaics with respect to XCI status, with an expected ratio of 1:1.

However, some individuals display a skewed pattern of XCI (XCI-skew), which is defined as a deviation from the expected 1:1 ratio. Examples of primary XCI-skew have been identified, including stochastic events resulting in XCI-skew across all tissues (*Tukiainen et al., 2016*) or the preferential selection of cells expressing functioning alleles in heterozygous females with X-linked recessive traits (*Busque and Gilliland, 1998*; *Nyhan et al., 1970*). However, secondary or age acquired XCI-skew is more common and refers to increasing XCI-skew with age, particularly in mitotically active blood tissue (*Busque et al., 1996*; *Gale et al., 1997*). Within individuals, the correlation of XCI ratios between blood and other tissues diminishes over the life course as the XCI ratios in blood continue to skew with age (*Bolduc et al., 2008*; *Zito et al., 2019*).

The stability of XCI-skew in blood has been demonstrated over 18–24 months and is thought to be a gradual process affecting the whole haematopoietic stem cell population rather than representing fluctuations in the active stem cell pool (*Tonon et al., 1998*; *van Dijk et al., 2002*). Therefore, though XCI-skew is a sex-specific measurable phenotype, it is a potential marker of stem cell depletion or polyclonal expansion of haematopoietic stem cells, which are age-associated traits irrespective of chromosomal sex (*Busque et al., 1996*; *Busque et al., 2012*; *Gale et al., 1997*). Age acquired XCI-skew has previously been linked to autoimmunity, which presents with a stark sex-imbalance (*Chabchoub et al., 2009*), as well as breast and ovarian cancers, albeit with inconsistent findings (*Kristiansen et al., 2002*; *Lose et al., 2008*; *Manoukian et al., 2013*; *Struewing et al., 2006*). Yet the consequences of XCI-skew on chronic disease risk in an unselected population have largely been unexplored.

Clonal expansion of haematopoietic stem cells is also measurable by somatic mutations shared across blood cells, indicating a common stem cell precursor (*Xie et al., 2014*). Clonal haematopoiesis of indeterminate potential (CHIP) is a cellular phenotype describing a pre-malignant state in which ≥4% of blood cells harbour the same somatic mutation (*Jaiswal and Ebert, 2019*), thus representing monoclonal expansion. CHIP is robustly associated with all-cause mortality (*Jaiswal et al., 2014*), haematological cancers (*Genovese et al., 2014*), and cardiovascular disease (*Jaiswal et al., 2017*). XCI-skew can sometimes be a marker of CHIP: XCI-skew was previously used to determine clonality (*Busque and Gilliland, 1998*), and exome sequencing of females with XCI-skew identified *TET2* mutations as drivers of pre-malignant clonal haematopoiesis. However, XCI-skew and CHIP are not completely mutually inclusive (*Busque et al., 2012*).

Given XCI-skew is potentially tagging changes to the haematopoietic stem cell pool, we hypothesised that XCI-skew may be a marker of biological ageing and a risk factor for chronic disease. We tested this hypothesis by assaying XCI-skew in 1575 females from the TwinsUK cohort and employed prospective, cross-sectional, and intra-twin study designs to characterise the relationship of XCI with molecular and cellular measures of ageing, cardiovascular disease risk, and cancer diagnoses.

## Methods

### TwinsUK cohort

Archival DNA samples derived from whole blood (collected 1997–2017) were selected from individuals of the TwinsUK population cohort (**Verdi et al., 2019**). Twin pairs were date matched and the final dataset of 1575 samples comprised 423 monozygotic (MZ) twin pairs ($n_{individuals}$ = 846), 257 dizygotic (DZ) twin pairs ($n_{individuals}$ = 514), and 215 singletons (**Figure 1—figure supplement 2**). The age range of the XCI cohort is 19–99, with a median age of 61 (Figure 2A). All samples and information were collected with written and signed informed consent, including consent to publish within the TwinsUK study. TwinsUK has received ethical approval associated with TwinsUK Biobank (19/NW/0187), TwinsUK (EC04/015) or Healthy Ageing Twin Study (HATS) (07 /H0802/84) studies from NHS Research Ethics Service Committees London – Westminster.

### Human Androgen Receptor Assay (HUMARA)

The HUMARA method combines methylation-sensitive restriction enzyme digest and amplification of a highly polymorphic (CAG)n repeat in the first exon of the X-linked *AR* gene, allowing for the differentiation of the active and inactive chromosomes in heterozygous individuals (**Cutler Allen et al., 1992**). Here, 625 ng of genomic DNA was divided into three aliquots and incubated for 30 min at 37 °C with (i) the methylation-sensitive enzyme *HpaII*, (ii) the methylation-insensitive enzyme *MspI*, or (iii) water (mock digest) in 1 × New England Biolabs CutSmart Buffer. The *HpaII* digest was followed by an additional 20 min at 80 °C to avoid residual enzymatic activity. Fluorescently labelled PCR primers (FAM, VIC, NED, or PET; Forward primer 5'-dye-GCTGTGAAGGTTGCTGTTCCTCAT-3', Reverse primer 5'-TCCAGAATCTGTTCCAGAGCGTGC-3') were used in New England BioLabs One *Taq* Master Mix to amplify 1.5 µl of digested PCR product. The Mock and *HpaII* digested DNA were amplified in triplicate (using FAM, VIC, and NED), and the *MspI* digest, used as control of digestion efficiency, was amplified once (using PET). All PCRs were amplified with an initial denaturation step at 94 °C for 5 min, followed by 30 cycles of 94 °C for 30 s, 60 °C for 1 min, and 72 °C for 2 min, and a final elongation step of 72 °C for 15 min. To minimize technical bias and batch effects, the labelled amplified products were diluted 1:15 with nuclease-free ddH$_2$O and pooled together with the GeneScan 500 LIZ size standard and analysed on an ABI 3730xl. Twin pairs were assayed on the same plate and plates contained a mix of both MZ and DZ pairs. Two replicates were included on each plate and a within-plate correlation of 0.99 was measured. A total of 2382 DNA samples were assayed, with 194 failed samples, and 601 samples were homozygous for the CAG repeat and were therefore uninformative (**Figure 1—figure supplement 2**).

### Calculation of XCI

Data were analysed using the Microsatellite Analysis Software available on the Thermo Fisher Cloud. The XCI status was calculated in each of the triplicates as follows:

- Allele Ratio Mock Digestion (Rm)=allele 1 peak height / allele 2 peak height
- Allele Ratio *HpaII* Digestion (Rh)=allele 1 peak height / allele 2 peak height
- Normalized Ratio (Rn)=Rh/Rm
- XCI percentage = [Rn/(Rn +1)] * 100

A coefficient of variation (CV) was calculated across the triplicates and samples with CV >0.15 were excluded from downstream analysis (n=12; **Figure 1—figure supplement 2**). A mean XCI percentage (0–100%) was calculated for each sample, where 50% is perfectly balanced XCI and the directionality of XCI away from 50% is uninformative (e.g., both 0% and 100% are considered equal). Therefore, the XCI values are collapsed to a range of 50–100% when XCI is the dependent variable in analyses.

### XCI-skew categorical variable

The XCI percentage data were normalised, and a categorical XCI-skew variable was created from the absolute values of the normalised distribution as follows: standard deviation (SD)<1 from the mean = random XCI (0); 1≤SD<2 = skewed XCI (1); and SD≥2 = extreme skew (2). As such, XCI-skew equated to ≥75% XCI, and extreme XCI-skew equated to ≥91% XCI (**Figure 1—figure supplement 1**). These thresholds are very similar to previous studies (**Busque et al., 1996**; **Gale et al., 1997**), and allowed for linear associations to be tested using the XCI-skew categorical variable.

## Statistical analysis

In all regression models, a linear mixed effects model was used with relatedness and family structure fitted with a random intercept using the lme4 package (*Bates et al., 2015*). The relevant fixed effects are described in each section below and were specific to each test. All analyses were carried out using R version 4.1.1, and all plots were generated using ggplot (*Wickham, 2016*).

## Chronological and biological ageing

Datasets were matched to be within 1 year of the XCI DNA sample and the significance threshold after Bonferroni correction was p<0.007 to account for multiple testing across the seven tests in this section. Chronological age was calculated at time of DNA sampling and the association was tested using XCI as the dependent variable. Body Mass Index (BMI) measures were taken during clinical visits and obesity was defined as BMI ≥30. Smoking status was classified based on longitudinal questionnaire answers (*Christiansen et al., 2021*). The associations were tested with XCI as the dependent variable and obesity (obese/not obese) or smoking (ever/never smoker) as the independent variable, controlling for age as a fixed effect. A Frailty Index (*Searle et al., 2008*) was calculated based on longitudinal questionnaire data and used as the dependent variable, with XCI-skew as the independent variable and age and BMI as fixed effects. Leukocyte Telomere Length was measured using qPCR as previously described (*Codd et al., 2010*), and the normalised measures were used as the dependent variable and XCI-skew as the independent variable, with age and smoking as fixed effects. DNA methylation (DNAm) GrimAge was calculated using 450 K methylation data and GrimAge epigenetic age acceleration measures were obtained from regressing epigenetic age on chronological age (*Costeira et al., 2021*). GrimAge Acceleration was used as the dependent variable and XCI-skew as the independent variable, with no additional covariates included in the model.

## Whole blood count data

Automated whole blood count data were date-matched to the XCI DNA sample, and each of the 10 blood count variables was normalised. The significance threshold after Bonferroni correction was p<0.005 to account for multiple testing across the 10 tests. In addition, Monocyte-to-Lymphocyte Ratio (MLR) and Neutrophil-to-Lymphocyte Ratio (NLR) were calculated by dividing the total monocyte or neutrophil count, respectively, by total lymphocyte count, and a Bonferroni-corrected significance threshold of $P<0.025$. Associations were tested with XCI-skew as an independent variable after controlling for age, BMI, seasonality, and smoking status as fixed effects in a linear mixed effects model.

## Cytokines levels and C-reactive protein

Serum IL-1β, IL-10, IL-6, and TNF were measured simultaneously using the bead-based high sensitivity human cytokine kit (HSCYTO-60SK, Linco-Millipore) according to the manufacturers' instructions. CRP concentrations from serum were measured with the Human Cardiovascular Disease Panel 2 LINCOplex Kit (HCVD2-67BK, Linco-Millipore) and with the Extracellular Protein Buffer Reagent Kit (LHB0001, Invitrogen). CRP concentrations were diluted 1:2000 prior to analysis and assayed in duplicate, as previously described (*Ligthart et al., 2018*). Date-matched data with batch effects regressed out were normalised and associations were assessed using linear mixed models controlling for seasonality and age as fixed effects. The significance threshold after Bonferroni correction was p<0.01 to account for multiple testing across five tests.

## Atherosclerosis and cardiovascular disease (ASCVD) risk score

The ASCVD risk score was calculated for a subset of 228 individuals with date-matched data on age, total cholesterol, HDL cholesterol, smoking, diabetes, systolic blood pressure, and hypertension medication, as previously described (*Goff et al., 2014*). A linear mixed effects model was used to control for BMI and monocyte abundance as fixed effects (*Madjid et al., 2004*). Twin pairs discordant for XCI-skew but matched for date of visit and age (n=34 pairs) were used for the intra-twin study, and ASCVD risk scores were compared using a one-sided paired samples Wilcoxon test.

## Cancer and all-cause mortality

Anonymised data were obtained from the National Disease Registration Service. Study entry was the date of DNA sampling and follow-up occurred through to January 2020 (study end date). For the cancer analysis, individuals who had not experienced the event were censored at 10 years, study end date, or date-of-death. All participants with a history of cancer before sampling, or within 6 months of sampling, were excluded from analyses, and reports of non-melanoma skin cancer were filtered, leaving a sample size of 1417. For all-cause mortality, individuals who had not experienced the event were censored at 10 years or study end date. For both analyses, age, relatedness, and zygosity were controlled for in the Cox regression model using R package Survival (*Therneau, 2021*). Proportional hazards assumptions were assessed using the cox.zph function of the Survival package. Kaplan-Meier plots were used for graphical representation of years until diagnosis. XCI-skew and extreme XCI-skew groups were combined due to the limited number of events.

## Results

### XCI-skew in the TwinsUK population cohort

We measured XCI in DNA derived from whole blood using the methylation-sensitive PCR-based Human Androgen Receptor Assay (HUMARA), which differentiates between alleles from the active and inactive X (*Cutler Allen et al., 1992*; *Hatakeyama et al., 2004*). HUMARA is an extensively used

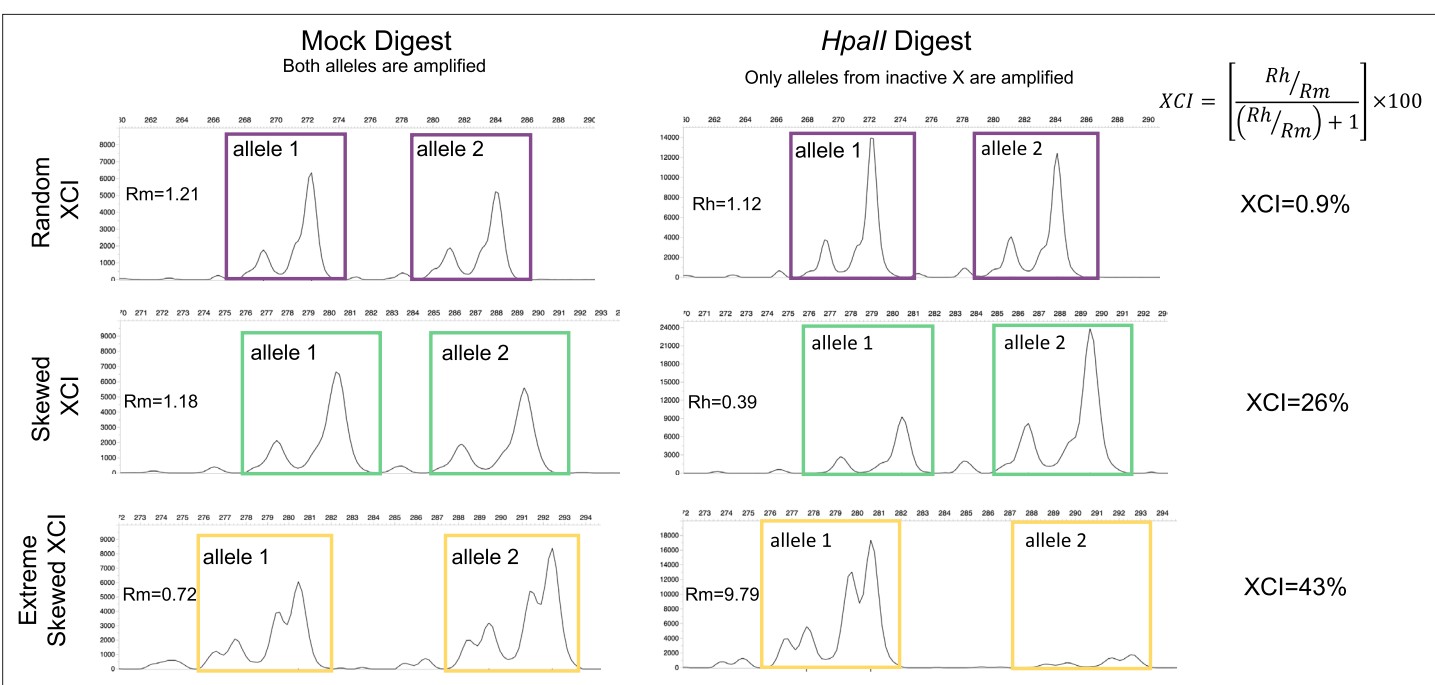

**Figure 1.** Measuring XCI with the HUMARA assay. The Human Androgen Receptor Assay (HUMARA) uses methylation-sensitive restriction enzyme digest and PCR to measure skewed X-inactivation. The assay estimates XCI-skew by comparing the relative abundance of allele specific fragments from a mock digest to a methylation-sensitive *HpaII* digest in which only the alleles from the inactive X are amplified. Representative examples are displayed of fragment analysis of the PCR products for samples with random XCI (top), skewed XCI (middle), and extreme skewed XCI (bottom). The x-axis shows the size, and the y-axis represents the abundance, of the PCR products, respectively. The left panel shows the PCR products after a mock digest with water, resulting in amplification of both alleles regardless of chromosomal inactivation. The right panel shows the PCR products after a restriction enzyme digest with methylation-sensitive enzyme *HpaII*, resulting in amplification of only the alleles deriving from inactive chromosomes. For each sample, the ratio of the *HpaII* digested allele products (Rh = allele 1/allele 2) is divided by the ratio of the Mock digest allele products (Rm = allele 1/ allele 2) to create a Normalized Ratio (Rn). The XCI percentage is then calculated using the formula [Rn/(Rn +1)] * 100. Images were generated using the Microsatellite Analysis Software on the Thermo Fisher Cloud.

The online version of this article includes the following figure supplement(s) for figure 1:

**Figure supplement 1.** A histogram showing the calculation of the XCI-skew categorical variable from the normalised distribution of XCI scores across the TwinsUK cohort (n=1575).

**Figure supplement 2.** A flowchart of sample processing and inclusion criteria for each results group.

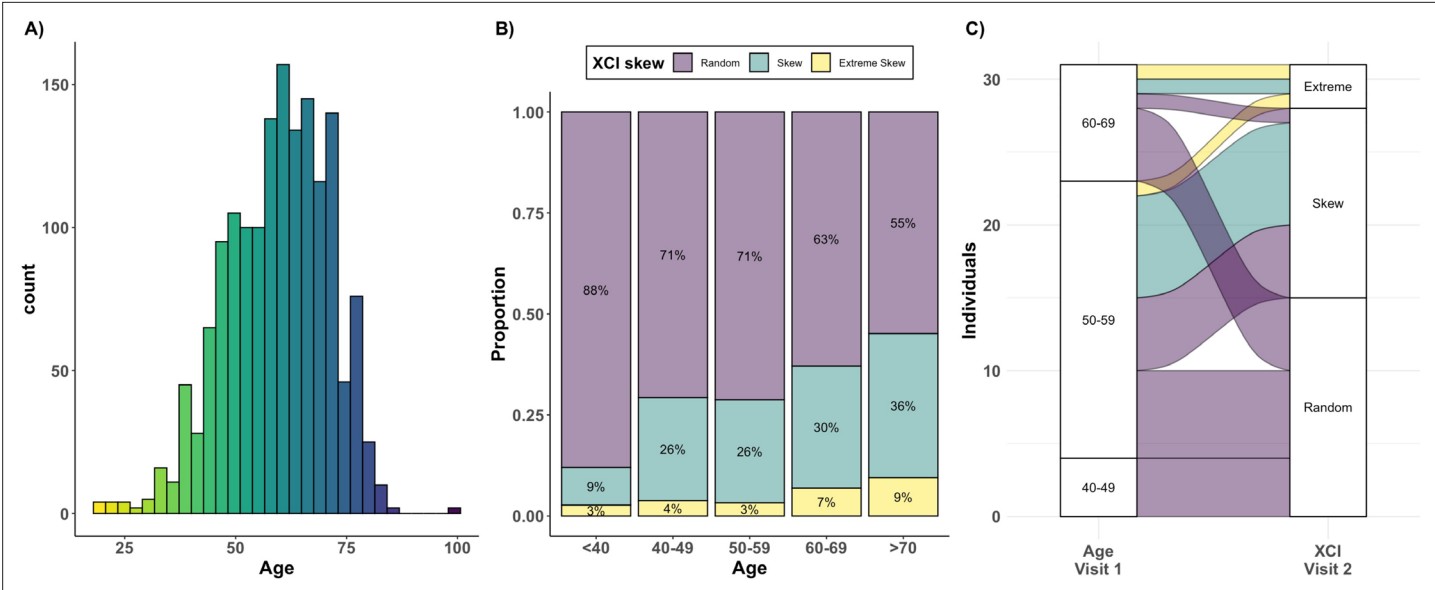

**Figure 2.** Age acquired XCI-skew across age groups and time. (**A**) A histogram displaying the age distribution of the TwinsUK HUMARA cohort (age range: 19–99; median age = 61). (**B**) The proportions of individuals (y-axis) in each of three XCI-skew categories across increasing age groups (x-axis) are shown (N=1575). (**C**) A Sankey plot shows the longitudinal changes to XCI in 31 individuals across two measures 15–17 years apart. Colours indicate XCI at visit 1, axis 1 displays the age group of individuals at visit 1, and axis 2 displays XCI at visit 2.

The online version of this article includes the following source data for figure 2:

**Source data 1.** XCI-skew across age groups.

assay which correlates well with transcription-based methods (**Bolduc et al., 2008**; **Zito et al., 2019**). The output of the HUMARA assay is a continuous XCI variable from 0–100%, where 50% is perfectly balanced XCI and the directionality of XCI away from 50% is uninformative (e.g., both 0% and 100% are considered equal). We normalised the distribution of the continuous XCI values across the cohort, and defined XCI-skew as measures 1 SD from the mean, corresponding to XCI score ≥75%, and extreme XCI-skew as measures 2 SD from the mean, corresponding to XCI score ≥91% (**Figure 1—figure supplement 1**). Of note, these values are extremely similar to thresholds previously used in the literature (**Bolduc et al., 2008**; **Busque and Gilliland, 1998**). **Figure 1** shows representative samples of how the HUMARA assay relates to the three categorical XCI-skew statuses.

We defined XCI-skew in 1575 participants (median age = 61; **Figure 2A**) unselected for chronic disease status from the TwinsUK population cohort (**Verdi et al., 2019**), which comprised of 423 MZ pairs, 257 DZ pairs, and 215 singletons. In line with previous studies (**Vickers et al., 2001**), we see increased concordance of XCI-skew within MZ twin pairs compared with DZ twin pairs: 27% of MZ twin pairs (114/423), and 45.5% of DZ twin pairs (117/257), were discordant for their categorical XCI-skew status. We date-matched the XCI data with existing phenotypes from TwinsUK (e.g., blood count data, molecular markers) in the subsets of individuals on whom each phenotype was available (**Figure 1—figure supplement 2**).

## Cross-sectional and longitudinal changes to XCI-skew with age

We assessed changes in frequency of XCI-skew across increasing age groups and identified 12% (9 of 75) of individuals under 40 years old (yrs) displaying XCI-skew (≥75% XCI); 28% (183 of 652) of 40–59 yrs; 37% (185 of 498) of 60–69 yrs; and 44% (132 of 303) of those over 70 yrs (**Figure 2B**). Proportions of individuals displaying extreme XCI-skew (≥91% XCI) remains consistent at ~3–4% below the age of 60 but increases to 7% of 60–69 yrs and 9% of those over 70 yrs. These results suggest a stepwise increase in prevalence of XCI-skew happening after 40 years of age, then again after 60 years of age, where we also see the first increase in prevalence of extreme XCI-skew (**Figure 2B**). As expected, after controlling for relatedness and zygosity, we find a significant positive association between age and XCI skewing (p=2.8 × 10⁻⁹, N=1575). This result replicates the many existing studies

on age acquired XCI-skew and acts as a validation of the TwinsUK HUMARA dataset (*Busque et al., 1996*; *Gale et al., 1997*; *Hatakeyama et al., 2004*; *Zito et al., 2019*).

We assessed change in XCI-skew over time using 31 individuals on whom we had an additional second sample available from 15 to 17 years prior to the main study (*Figure 2C*. median age at visit 1=55.5; median age at visit 2=72.1). The two individuals who had extreme XCI-skew at visit 1 still displayed extreme XCI-skew at visit 2. Of the eight individuals who had XCI-skew at visit 1, seven remained skewed and one progressed to extreme XCI-skew at visit 2. Of the 21 who had a random pattern of XCI at visit 1, 15 (71.4%) remained the same, and 6 (28.6%) progressed to XCI-skew at visit 2. These longitudinal data indicate that XCI-skew categorisation persists over extended periods of time and increases over the life course.

## XCI-skew is independent of known markers of biological ageing

Ageing is a heterogenous process in which an individual's biological age can differ from their chronological age. Smoking and obesity are risk factors for accelerated ageing (*Tam et al., 2020*). Accelerated ageing can be estimated through measures of frailty (*Clegg et al., 2013*; *Chen et al., 2014*),

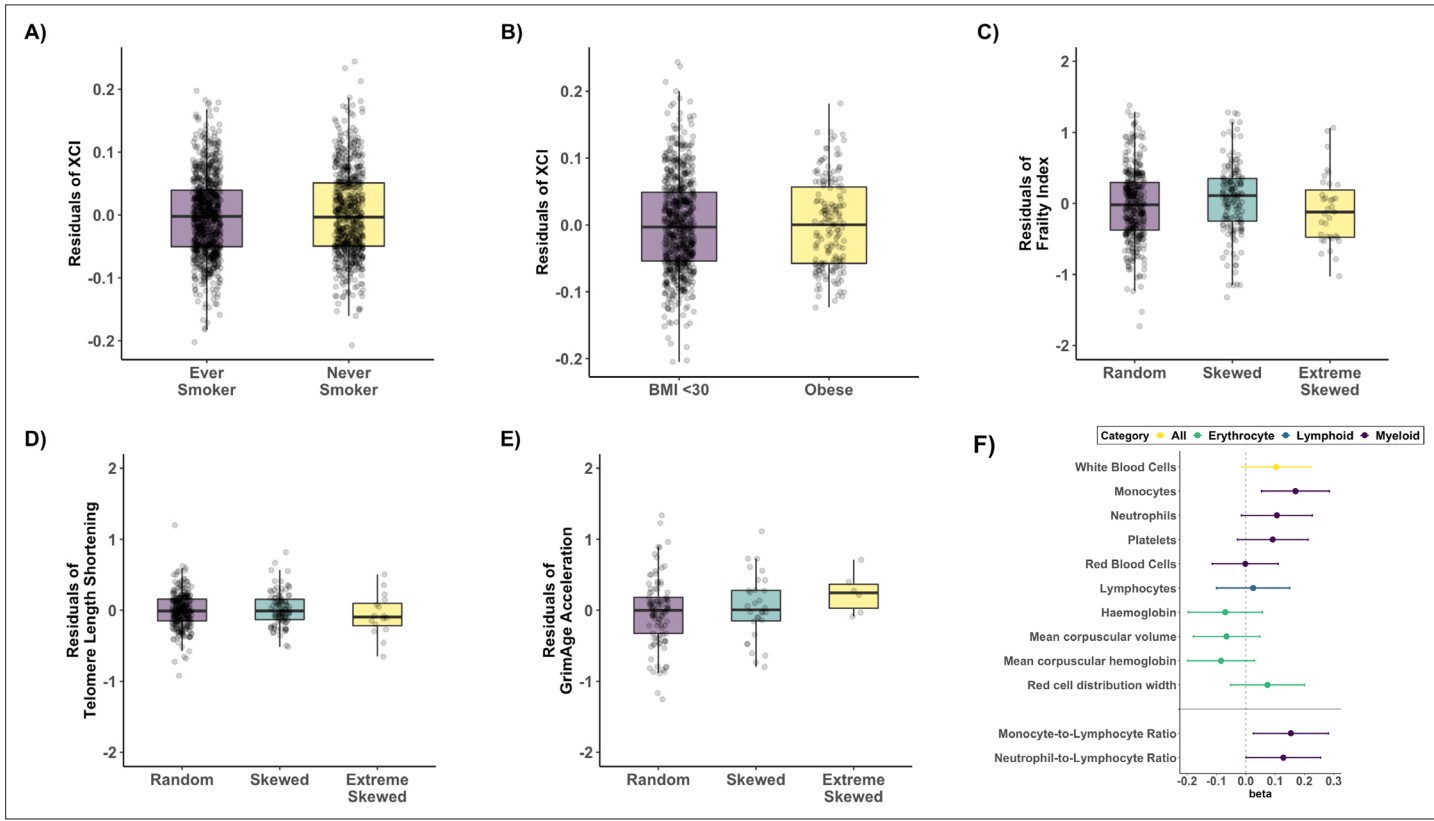

**Figure 3.** Age acquired XCI-skew and markers of biological ageing and blood cell counts. Box plots representing the results of the linear regression mixed effects models to assess (**A**) Smoking status (p=0.33, $N_{never\_smoker}$ = 879; $N_{ever\_smoker}$ = 673) and (**B**) obesity (p=0.88, $N_{not\_obese}$ = 726; $N_{obese}$ = 165) after correcting for age with XCI as the dependent variable, and (**C**) frailty index (p=0.59, $N_{ranodm\ XCI}$ = 398; $n_{Skewed\ XCI}$ = 177; $N_{extreme\ skew}$ = 36), after correcting for age and BMI, (**D**) Leukocyte telomere length shortening (p=0.9, $N_{ranodm\ XCI}$ = 278; $n_{Skewed\ XCI}$ = 103; $N_{extreme\ skew}$ = 16) after correcting for age and smoking status, and (**E**) DNAm GrimAge acceleration (p=0.22, $N_{ranodm\ XCI}$ = 101; $n_{Skewed\ XCI}$ = 30; $N_{extreme\ skew}$ = 6), with XCI-skew as the dependent variable. All boxplots display the median and IQR, and have the residuals of the models on the y-axis. (**F**) A forest plot of associations with data-matched Complete Blood Count data (top panel) and Myeloid-to-lymphoid ratios (bottom) with effect size and lower and upper confidence intervals indicated. Associations were tested with XCI-skew as an independent variable after controlling for age, BMI, seasonality, and smoking status as fixed effects in a linear mixed effects model ($N_{ranodm\ XCI}$ = 445; $n_{Skewed\ XCI}$ = 183; $N_{extreme\ skew}$ = 43). The significance threshold after Bonferroni correction was p<0.005 and p<0.023 to account for multiple testing across the 10 tests and 2 tests, respectively.

The online version of this article includes the following source data and figure supplement(s) for figure 3:

**Source data 1.** XCI-skew and DNAm GrimAge Acceleration.

**Figure supplement 1.** Age acquired XCI-skew and Cytokine and CRP measures.

and on a molecular level using measures of leukocyte telomere length shortening (*Blackburn et al., 2006*) and epigenetic ageing clocks such as DNA methylation (DNAm) GrimAge (*Lu et al., 2019*); all these measures are associated with adverse health outcomes (*Blackburn et al., 2006*; *Hewitt et al., 2020*; *Lu et al., 2019*).

Given the robust association with chronological age, we sought to establish whether XCI-skew was associated with biological ageing using measures taken within 1 year of the XCI DNA sample and using linear regression mixed effects models, controlling for relatedness and zygosity as random effects (*Figure 3*). We observed no association with smoking status (p=0.33, $N_{never\_smoker}$=879; $N_{ever\_smoker}$ = 673; *Figure 3A*), nor obesity (p=0.88, $N_{not\_obese}$=726; $N_{obese}$ = 165; *Figure 3B*), after correcting for age. We also found no association with a robust frailty index (p=0.59, $N_{ranodm\ XCI}$=398; $n_{Skewed\ XCI}$ = 177; $N_{extreme\ skew}$ = 36; *Figure 3C*) after correcting for age and BMI, nor with leukocyte telomere length shortening (p=0.9, $N_{ranodm\ XCI}$=278; $n_{Skewed\ XCI}$ = 103; $N_{extreme\ skew}$ = 16; *Figure 3D*), after correcting for age and smoking status. Finally, we see no association with DNAm GrimAge acceleration (p=0.22, $N_{ranodm\ XCI}$=101; $n_{Skewed\ XCI}$ = 30; $N_{extreme\ skew}$ = 6; *Figure 3E*), however, we believe the relationship between XCI-skew and epigenetic ageing would benefit from a follow-up study with a larger sample size particularly given the limited number of samples with extreme XCI-skew in our analysis. Together, these data, ranging from the molecular to organismal level, suggest age acquired XCI-skew is independent of many known markers of biological ageing and is potentially a unique biomarker with unexplored utility.

## XCI-skew is associated with increased monocyte abundance and decreased IL-10 levels

Changes in blood cell composition can be indicative of ill health or systemic inflammation (*Kabat et al., 2017*; *Madjid et al., 2004*; *Patel et al., 2009*), and a haematopoietic stem cell bias towards the myeloid lineage is observed with ageing (*Pang et al., 2011*). We tested for associations between XCI-skew and whole blood count data, including white cell differentials, in a subset of individuals with matched data ($N_{ranodm\ XCI}$ = 445; $n_{Skewed\ XCI}$ = 183; $N_{extreme\ skew}$ = 43, median age = 63). After controlling for age, seasonality, BMI, smoking, relatedness and zygosity in a linear regression mixed effects model, XCI-skew is association with increased monocyte abundance after multiple testing correction (p=0.0038), and we observed nominal increases in abundance across other myeloid cells (*Figure 3F*). We next tested the hypothesis that XCI-skew was associated with a myeloid lineage bias using the Monocyte-to-Lymphocyte Ratio (MLR) (*Chen et al., 2019*) and Neutrophil-to-Lymphocyte Ratio (NLR) (*Arbel et al., 2012*), and detect an association between XCI-skew and MLR (p=0.019), and a nominal association with NLR (p=0.042) (*Figure 3F*). Though myeloid cells show a greater degree of skewing, thought to be due to the shorter lifespan of these cells (*Gale et al., 1997*), we do not believe the association seen here between XCI-skew and increasing monocyte numbers is causal: XCI-skew is defined as ≥25% shift in cell mosaicism, whereas monocytes account for only ~10% of white blood cells.

'Inflammageing' is the chronic pro-inflammatory phenotype observed in ageing and is considered an altered state of intercellular communication (*López-Otín et al., 2013*). Markers of inflammageing include cytokines produced by immune cells and C-reactive protein (CRP) produced by liver cells (*Salminen et al., 2012*). We date-matched the XCI data with serum levels of CRP ($N_{ranodm\ XCI}$ = 121; $n_{Skewed\ XCI}$ = 38; $N_{extreme\ skew}$ = 6) and a more modest cytokine dataset of interleukin (IL)–6, IL-1B, IL-10, and TNF ($N_{ranodm\ XCI}$ = 23; $n_{Skewed\ XCI}$ = 4) and used linear regression mixed effects models to control for age, seasonality, relatedness and zygosity (*Figure 3—figure supplement 1*). We see no association with primary markers of inflammageing CRP (p=0.41), IL-6 (p=0.41), or TNF (p=0.61), though a nominal association with IL-1β is observed (p=0.02) which does not pass multiple correction, but warrants follow up analysis with a larger sample size. However, we observe a strong negative association with IL-10 (p=0.0008, *Figure 3—figure supplement 1*). IL-10 is a broadly expressed anti-inflammatory cytokine which can inhibit the proinflammatory responses of both innate and adaptive immune cells (*Saraiva and O'Garra, 2010*). Though we note that due to minimal overlap in the datasets, we were unable to control for cell type composition, which may impact these findings.

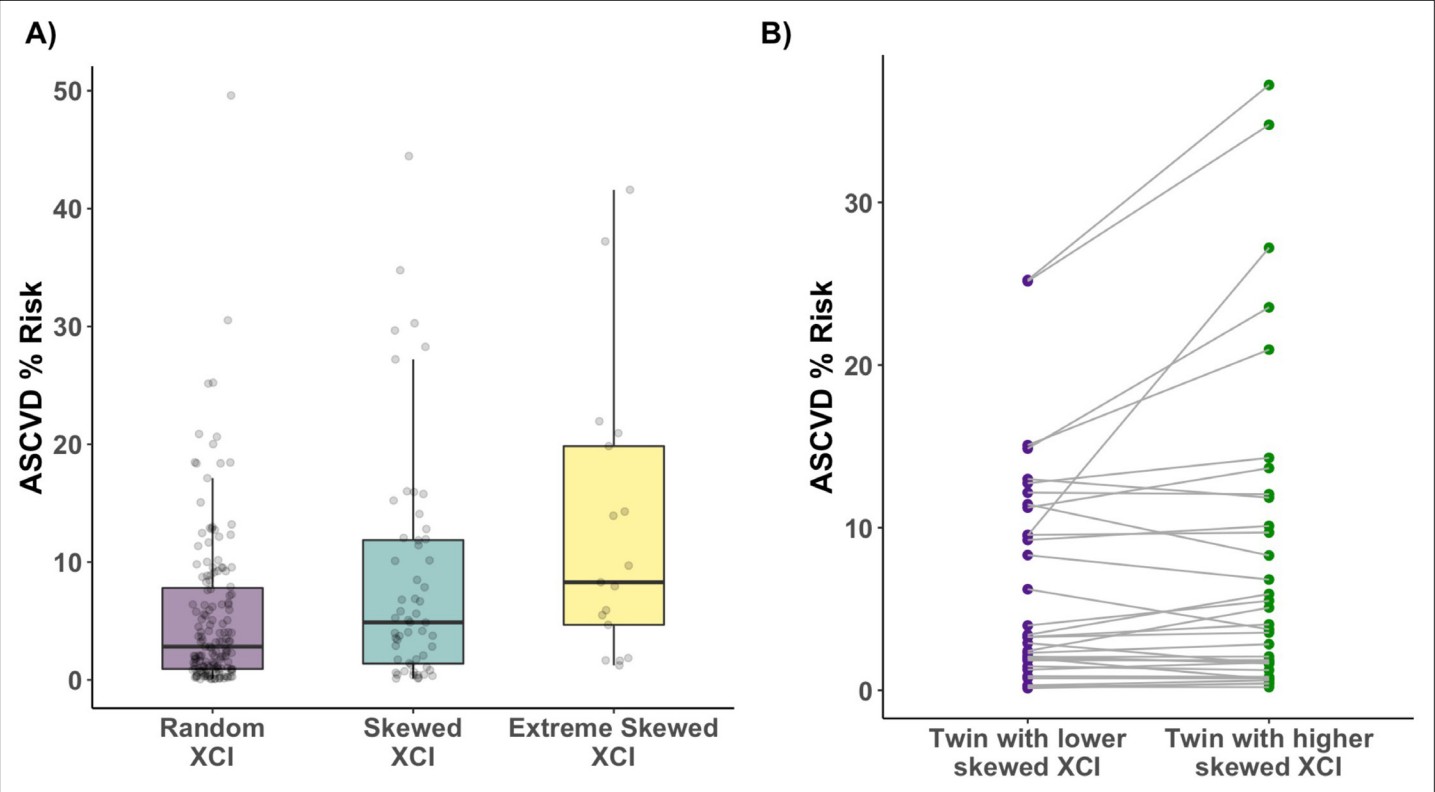

**Figure 4.** Age acquired XCI-skew and cardiovascular disease risk score. (**A**) Using a linear regression mixed effects model to control for BMI, monocyte count, relatedness and zygosity, XCI-skew is associated with increased atherosclerotic cardiovascular disease risk score ($N_{ranodm\ XCI}$ = 155; $n_{Skewed\ XCI}$ = 56; $N_{extreme\ skew}$ = 17; p=0.01). The boxplot displays the median and IQR. (**B**) Using age-matched twin pairs discordant for their XCI-skew status ($N_{pairs}$ = 34), XCI-skew is associated with increased atherosclerotic cardiovascular disease risk score in the intra-twin analysis (one-sided paired samples Wilcoxon test; p=0.025).

The online version of this article includes the following source data for figure 4:

**Source data 1.** XCI-skew and ASCVD Risk Score.

## Atherosclerotic cardiovascular disease risk is increased in individuals with XCI-skew

Cardiovascular disease (CVD) is the leading cause of death worldwide, and monocytes are an innate immune cell type known to be mediators in CVD disease progression and are found in atherosclerotic lesions (*Libby et al., 2016*). Given the association of XCI-skew with increased monocyte abundance, and the association of CHIP with CVD, we hypothesised that XCI-skew could also be associated with CVD. The atherosclerotic cardiovascular disease (ASCVD) risk score (*Goff et al., 2014*) (see Methods) captures traditional risk factors and gives a predicted risk of a major CVD event in the next 10 years, with an ASCVD risk score >7.5% representing intermediate risk, and an ASCVD risk score >20% representing high risk. In a cross-sectional study of 228 individuals ($N_{ranodm\ XCI}$ = 155; $n_{Skewed\ XCI}$ = 56; $N_{extreme\ skew}$ = 17; median age = 62) with matched health data available, XCI-skew was associated with increased ASCVD risk score after controlling for BMI, monocyte abundance, relatedness and zygosity using a linear regression mixed effects model (p=0.01, *Figure 4A*). 23.5% (4 of 17) and 35.3% (6 of 17) of individuals with extreme XCI-skew have high and intermediate ASCVD risk, respectively, compared to 4.5% (7 of 155) and 21.9% (34 of 155) of individuals with random XCI.

To ensure the observed association wasn't spuriously driven by the age component of the ASCVD risk score (see Methods), we took age matched MZ and DZ twin pairs ($n_{pairs}$ = 34) discordant for their XCI-skew status and tested intra-twin ASCVD risk scores. The intra-twin analysis validated the association between higher XCI-skew and increased ASCVD risk score (one-sided paired samples Wilcoxon test p=0.025; *Figure 4B*), adding further support to the finding.

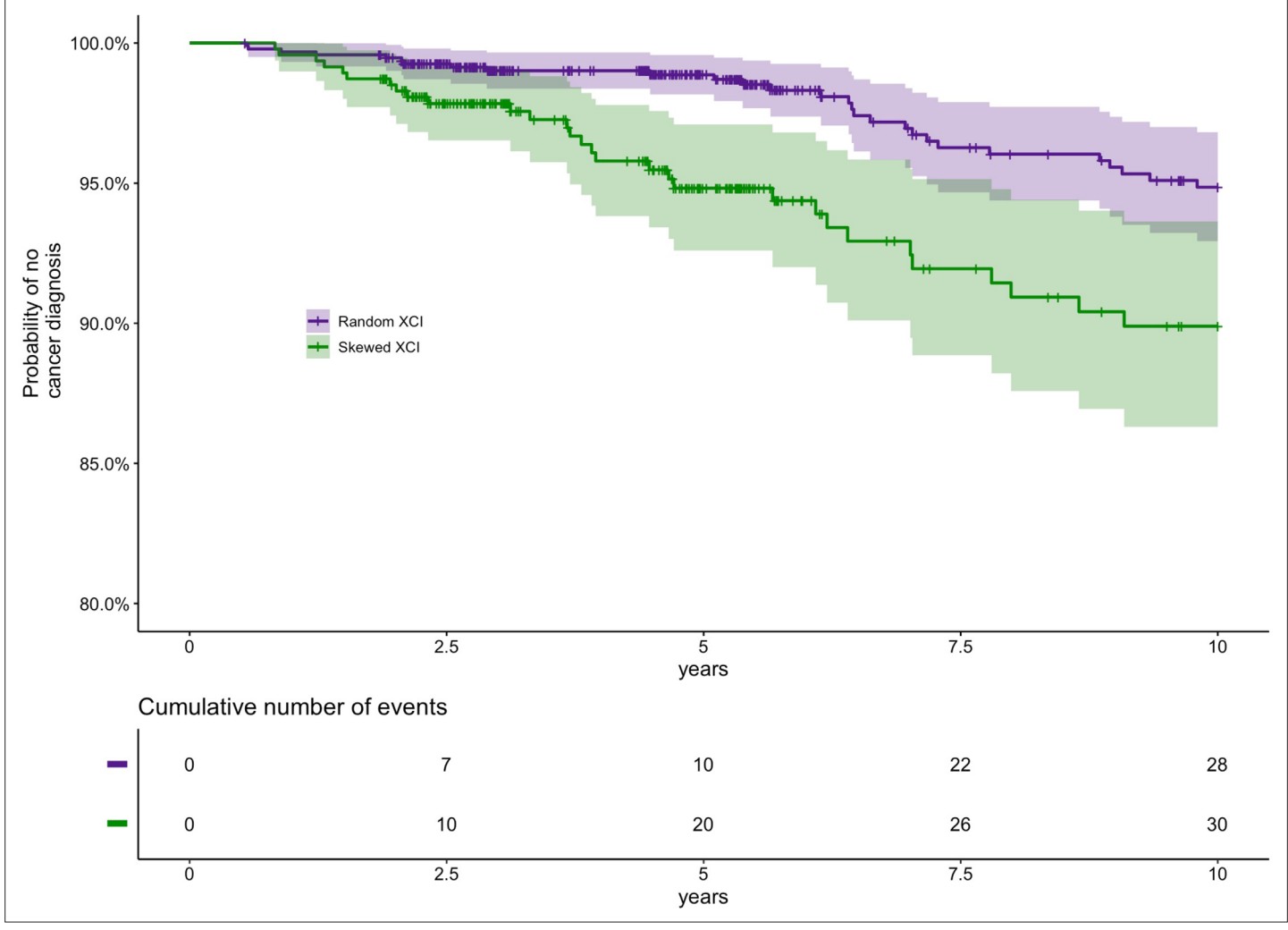

**Figure 5.** Prospective study of XCI-skew and future cancer diagnosis. Using a Cox regression analysis and controlling for age, relatedness and zygosity, XCI-skew is predictive of future cancer incidence in a 10 year follow-up (p=0.012; HR = 1.95). Each cancer event represents the first cancer diagnosis (excluding non-melanoma skin cancer) in each individual. A Kaplan-Meier plot (top) and cumulative events (bottom) of cancer diagnosis in individuals with XCI-skew (N=469) and random XCI (N=948) are shown. 2.9% (28/948) of individuals with random XCI, and 6.4% (30/469) of individuals with XCI-skew, developed cancer in the 10 years follow-up.

## XCI-skew is predictive of future cancer diagnosis in 10-year follow-up

The association of XCI-skew with cancer has largely been assessed in case-control studies focusing on cancers of female reproductive organs, with limited replicated findings (*Buller et al., 1999*; *Kristiansen et al., 2002*; *Struewing et al., 2006*). We conducted a prospective 10 year follow-up study (median follow-up time 5.65 years) from time of DNA sampling in 1417 individuals ($N_{random}$ = 948, $N_{skewed}$ = 469, median age = 60) who were cancer-free at baseline to assess the association between XCI-skew and future cancer diagnoses (cancer events = 58; *Supplementary file 1*). Each cancer event represents the first cancer diagnosis (excluding non-melanoma skin cancer) in each individual and subsequent diagnoses are not included. Using multivariate Cox regression analysis controlling for age, relatedness and zygosity, XCI-skew was associated with increased probability of cancer diagnosis (p=0.012; Hazard Ratio (HR)=1.95 (95% Confidence Interval (CI)=1.16–3.28); *Figure 5*). Though we were underpowered to run associations for specific cancers, we were interested to explore the potential that blood cancers were partly driving the association given the robust association of CHIP with haematological cancers. However, running the analysis without the Haematopoietic/Lymphoid Tissue cancers (see *Supplementary file 1*) strengthened the observed association (p=0.009; HR = 2.04 [1.20–3.49]) suggesting a crucial difference in cancer risk between XCI-skew and CHIP.

It has been suggested that XCI-skew could have a protective effect against all-cause mortality in cohorts selected for longevity (*Mengel-From et al., 2012*). However, in our 10 year follow-up study in a population cohort (median follow-up time = 5.71 years; deaths = 41), we see a non-statistically significant trend towards a positive association with all-cause mortality (p=0.29; HR = 1.39 [0.75–2.58]).

## Discussion

The age association of XCI-skew in blood tissue has long been established, with increased prevalence of females displaying XCI-skew after middle age, which is a critical time for the development of chronic disease (*Busque et al., 1996*; *Gale et al., 1997*; *Zito et al., 2019*). Although XCI-skew is measurable in a sex-specific manner, age acquired XCI-skew is potentially a marker of stem cell depletion or clonal expansion of haematopoietic stem cells (*Busque et al., 1996*; *Busque et al., 2012*; *Gale et al., 1997*; *van Dijk et al., 2002*), and could therefore have a role in age-related chronic disease, which has been robustly established for CHIP (*Jaiswal and Ebert, 2019*). In our cross-sectional study of 1575 females, we identify associations of blood-derived age acquired XCI-skew with an increased atherosclerotic cardiovascular disease risk score and increased probability of future cancer diagnosis.

Intriguingly, XCI-skew appears independent of other known markers of biological ageing, including leukocyte telomere length shortening, inflammageing, and a robust measure of frailty, making XCI-skew a unique biomarker with clinical potential. We note that though we also see no association with the mDNA GrimAge measure of accelerated epigenetic ageing, our sample size was limited, and this perhaps warrants a follow-up study to better establish the lack of correlation. Importantly, we also demonstrate that smoking and obesity are not risk factors for age acquired XCI-skew, and with a limited longitudinal dataset, that XCI-skew persists and increases over extended periods of time (median 16 years).

We find an association with increased probability of future cancer diagnosis in a 10 year follow-up study. Previous studies assessing the role of XCI-skew in cancer have typically used case-control studies of breast and ovarian cancers and have not been robustly replicated. The lack of replication is potentially explained by the heterogeneity in study design, including age-of-onset and therapy timing (*Kristiansen et al., 2002*; *Struewing et al., 2006*), *BRCA1* mutation stratification (*Kristiansen et al., 2005*; *Lose et al., 2008*; *Manoukian et al., 2013*), and threshold level for definition of XCI-skew (*Buller et al., 1999*). We controlled for potential confounding effects of cancer treatment on blood cells by excluding all individuals with a cancer diagnosis prior to study entry. Due to limited cancer events, our Cox regression analysis combined all individuals with XCI-skew (≥75% XCI) and extreme XCI-skew (≥91% XCI), which suggests even modest levels of skewing represents a greater risk of cancer.

Though CHIP is also predictive of future cancer diagnosis, the association is limited to haematological cancers (*Desai et al., 2018*). Whereas here we see blood-derived measures of XCI-skew is predictive of all future cancer diagnoses, even when the haematological cancers are removed from the analysis. Follow-up studies to assess the risk of cancer of specific tissues in individuals with XCI-skew are needed, but we hypothesise that the relationship between XCI-skew measures in blood tissue and cancer will not be causal. Instead XCI-skew is likely a marker of chronic inflammation, which can predispose to the development of cancer through increased mutagenesis and can promote tumorigenesis by shaping the tumour microenvironment to stimulate tumour growth (*Greten and Grivennikov, 2019*). However, it is interesting that common environmental factors that induce chronic inflammation, such as smoking and obesity, have not been found to be risk factors for XCI-skew in our study.

We also present an association with increased atherosclerotic cardiovascular disease risk score, which captures traditional risk factors (age, total cholesterol, HDL cholesterol, smoking, diabetes, systolic blood pressure, and hypertension medication) and estimates the risk of developing CVD in the next 10 years (*Goff et al., 2014*). We were also able to utilise the powerful discordant twin design to validate this finding, thus excluding the possibility that the cross-sectional association was driven by the age component of the ASCVD risk score. Following our finding of a correlation between blood-derived XCI-skew and risk of CVD, ascertaining whether XCI-skew is associated with incident CVD is of upmost importance. CHIP has been robustly associated with incident CVD and this association is thought to be causal (*Jaiswal et al., 2017*). Furthermore, the association of CHIP and all-cause mortality is partly driven by CVD (*Jaiswal et al., 2014*). We lacked statistical power to demonstrate an association with all-cause mortality, though the trend was towards greater risk (HR = 1.39 [0.75–2.58]).

A higher-powered study able to focus on specific causes of mortality will be of great interest and importance.

Given the existing links between CHIP and XCI-skew as two age acquired blood traits, and that CHIP mutations can be found in individuals with XCI-skew (*Busque et al., 2012*), could the association between XCI-skew and CVD risk be partly driven by CHIP? It is expected that some of the individuals with XCI-skew in our study will also harbour CHIP mutations. However, a better understanding of the co-occurrence of XCI-skew and CHIP within individuals, and the mutational burden within individuals with and without XCI-skew, is an important area of future work. Whether the co-occurrence of XCI-skew and CHIP represents an amplified risk of disease, or helps better define risk categories, will be of clinical significance to establish. As the inherited genetic risk of CHIP is fast becoming better understood, deciphering the genetic predisposition to XCI-skew will also enable the assessment of potential shared genetic susceptibility between the two traits (*Kar et al., 2022*). However, many of the negative results in our study are crucial findings given they expose differences between the risk factors of XCI-skew and CHIP. Notably, recent work has demonstrated smoking and telomere length are causal risk factors for CHIP (*Kar et al., 2022*), whereas we see no association with either trait in our study, suggesting CHIP and XCI-skew have some distinct aetiologies.

On a cellular level, we observe that increased abundance of monocytes correlates with increased XCI-skew, which is of particular interest given that monocytes/macrophages are involved in the inflammatory pathophysiology of CVD (*Libby et al., 2016*). It is important to note however that changes in monocyte counts alone do not explain the observed levels of XCI-skew. Monocytes account for ~10% of white blood cells whereas XCI-skew is defined here as ≥25% shift in cell mosaicism. Furthermore, age acquired XCI-skew has previously been shown across isolated neutrophils, monocytes, and T cells, with correlations between these fractions (*Tonon et al., 1998*; *van Dijk et al., 2002*), albeit with lower levels of skewing in lymphoid cells (*Gale et al., 1997*). Instead, XCI-skew is likely associated with the age-related haematopoietic bias toward the myeloid lineage (*Pang et al., 2011*), as we see nominal associations across other myeloid cell types in addition to the monocyte- and neutrophil-to-lymphocyte ratios. Our study also demonstrates an association of reduced levels of IL-10 with increased XCI-skew. IL-10 is an anti-inflammatory cytokine produced by a broad range of immune cells (*Saraiva and O'Garra, 2010*) and subsets of monocytes differ in their capacities to secrete IL-10 (*Skrzeczyńska-Moncznik et al., 2008*). Follow-up work on inflammatory profiles linked to XCI-skew may reveal mechanistic insights.

We derived our threshold for XCI-skew from the normalised distribution of the continuous XCI values across the cohort, and defined XCI-skew as measures 1 SD from the mean, corresponding to XCI score ≥75%, and extreme XCI-skew as measures 2 SD from the mean, corresponding to XCI score ≥91%. These values are very similar to thresholds used in the literature (*Busque et al., 1996*; *Gale et al., 1997*; *Zito et al., 2019*), but allow us to test for linear associations across the increasing thresholds of XCI-skew and demonstrate that individuals with lower levels of XCI-skew, which affects >37% of females over 60, are still at elevated risk of ASCVD and future cancer diagnoses. However, to avoid Type I error inflation, we did not run additional tests to compare the extreme XCI-skew group with the random XCI group.

There are some limitations to our study. Despite the high sample size of individuals with measured XCI, analyses are carried out in subsets of these individuals where date-matched phenotype data were available. In particular, the cytokine analyses have a low sample size and though we detect significant associations with IL-10, we may be underpowered to rule out the possibility of detecting a weaker association with other cytokines, particularly IL-1β. We were also unable to control for cell type composition in the cytokine analyses due to minimal overlap with the date-matched whole blood count data, which may have a significant impact on these findings. Though most of the measures of biological ageing were well-powered, the mDNA GrimAge analysis may benefit from replication in a larger sample size to more robustly establish the relationship between XCI-skew and accelerated epigenetic ageing. Also, with only 41 deaths across the cohort, our all-cause mortality analysis is underpowered, and a study able to focus on specific causes of mortality is warranted.

In summary, we demonstrate XCI-skew is a highly prevalent cellular phenotype in females and is associated with elevated cardiovascular disease risk and predictive of future cancer incidence. Further investigations are needed to translate the biological value of XCI-skew into clinical applications for studying the association of age-related haematopoietic changes and chronic disease associations,

regardless of chromosomal sex. Understanding the mechanisms underlying this phenomenon, whether XCI-skew is reflective of other ageing markers that increase disease risk, and whether it is therapeutically actionable, are areas of particular interest.

## Acknowledgements

The authors acknowledge use of the research computing facility at King's College London, *Rosalind* (https://rosalind.kcl.ac.uk), which is delivered in partnership with the National Institute for Health Research (NIHR) Biomedical Research Centres at South London & Maudsley and Guy's & St. Thomas' NHS Foundation Trusts, and part-funded by capital equipment grants from the Maudsley Charity (award 980) and Guy's & St. Thomas' Charity (TR130505). This work uses data that has been provided by patients and collected by the NHS as part of their care and support. The data are collated, maintained and quality assured by the National Disease Registration Service, which is part of NHS Digital. KSS acknowledges funding from the Medical Research Council [MR/M004422/1 and MR/R023131/1]. JTB acknowledges funding from the ESRC [ES/N000404/1]. MM acknowledges funding from the National Institute for Health Research (NIHR)-funded BioResource, Clinical Research Facility and Biomedical Research Centre based at Guy's and St Thomas' NHS Foundation Trust in partnership with King's College London. TwinsUK is funded by the Wellcome Trust, Medical Research Council, European Union, Chronic Disease Research Foundation (CDRF), Zoe Global Ltd and the National Institute for Health Research (NIHR)-funded BioResource, Clinical Research Facility and Biomedical Research Centre based at Guy's and St Thomas' NHS Foundation Trust in partnership with King's College London.

## Additional information

### Competing interests

Claire J Steves: Consultant for Zoe Ltd. The other authors declare that no competing interests exist.

### Funding

| Funder | Grant reference number | Author |
|---|---|---|
| Medical Research Council | MR/M004422/1 | Kerrin S Small |
| Medical Research Council | MR/R023131/1 | Kerrin S Small |
| Economic and Social Research Council | ES/N000404/1 | Jordana T Bell |
| National Institute for Health Research | | Massimo Mangino |

The funders had no role in study design, data collection and interpretation, or the decision to submit the work for publication.

### Author contributions

Amy L Roberts, Data curation, Formal analysis, Supervision, Investigation, Visualization, Methodology, Writing - original draft, Project administration; Alessandro Morea, Data curation, Investigation, Writing – review and editing; Ariella Amar, Investigation; Antonino Zito, Julia S El-Sayed Moustafa, Xinyuan Zhang, Data curation, Writing – review and editing; Max Tomlinson, Ruth CE Bowyer, Colette Christiansen, Ricardo Costeira, Data curation, Formal analysis, Writing – review and editing; Claire J Steves, Jordana T Bell, Resources, Writing – review and editing; Massimo Mangino, Resources, Data curation, Writing – review and editing; Chloe CY Wong, Supervision, Methodology, Writing – review and editing; Timothy J Vyse, Resources, Supervision, Writing – review and editing; Kerrin S Small, Conceptualization, Resources, Supervision, Funding acquisition, Project administration, Writing – review and editing

### Author ORCIDs

Amy L Roberts ⓘ http://orcid.org/0000-0002-5704-9249

Alessandro Morea http://orcid.org/0000-0002-7919-1932
Antonino Zito http://orcid.org/0000-0003-1931-984X
Claire J Steves http://orcid.org/0000-0002-4910-0489
Massimo Mangino http://orcid.org/0000-0002-2167-7470
Timothy J Vyse http://orcid.org/0000-0003-1123-1464
Kerrin S Small http://orcid.org/0000-0003-4566-0005

## Ethics

Human subjects: All samples and information were collected with written and signed informed consent, including consent to publish within the TwinsUK study. TwinsUK has received ethical approval associated with TwinsUK Biobank (19/NW/0187), TwinsUK (EC04/015) or Healthy Ageing Twin Study (HATS) (07/H0802/84) studies from NHS Research Ethics Service Committees London - Westminster.

## Decision letter and Author response

Decision letter https://doi.org/10.7554/eLife.78263.sa1
Author response https://doi.org/10.7554/eLife.78263.sa2

## Additional files

### Supplementary files
- Transparent reporting form
- Supplementary file 1. Number of cancer diagnoses recorded in 10-year follow-up by organ/site.
- Reporting standard 1. STROBE statement.

### Data availability

Source data are provided for Figure 2B, Figure 3E and Figure 4A. All data in the manuscript have been deposited to the TwinsUK BioResource data management team and are available by application to the Twin Research Executive Access committee (TREC) at King's College London. The TwinsUK BioResource is managed by TREC, which provides governance of access to TwinsUK data and samples. This excludes the National Disease Registration Service data which are only available to internal departmental members who are ONS accredited due to the terms of the data linkage. TwinsUK data users are bound by data sharing agreement set out in the data access application form (https://twinsuk.ac.uk/wp-content/uploads/2018/11/DTR_DataAccess_Policy_0318.pdf). This includes responsibilities with respect to third party data sharing and maintaining participant privacy. Further responsibilities include a responsibility to acknowledge data sharing.

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
