## [Editor Report]

XCI skewing is affected by age but how this may affect a person's health is not known. Roberts et al., demonstrate that these changes result in an increased risk of cardiovascular disease and cancer. These findings will be of interest to researchers studying the impact of age on health.

---

## [Decision Letter]

**Decision letter after peer review:**

Thank you for submitting your article "Age acquired skewed X Chromosome Inactivation is associated with adverse health outcomes in humans" for consideration by *eLife*. Your article has been reviewed by 3 peer reviewers, and the evaluation has been overseen by a Reviewing Editor and Carlos Isales as the Senior Editor. The following individual involved in the review of your submission has agreed to reveal their identity: Lambert Busque (Reviewer #1).

Essential revisions:

1. Was XCI measured on whole blood, versus fraction (PMN, Mononuclear cells, buffy coat, other)?

2. Did the authors look at the correlation between skewing in MZG vs DZ twins. This is known but would be interesting to document.

3. Was there enough discordant MZG twin pairs (best controls) for the ASCV risk comparison?

4. Age is the major confounding factor in all analyses. Although the information is present in the text and methods, I think controlling for age (or age-matched) should also be indicated in all figures.

5. The hypothesis that cardiovascular and cancer risk could be driven in part by CHIP should be added to the discussion.

6. The XCI skewing data is gathered using the HUMARA test. This is a classical test to score skewing in human cells. Since it is the central method of this manuscript, illustrative examples of the data (random, skew, and extreme skew, as well as non-informative samples) should be provided. This will be helpful to the reader and of broad interest to the *eLife* audience, especially for (epi)geneticists.

7. When mentioning karyotype, the most common nomenclature is 46, XX, and 46, XY instead of XX,46 or XY,46 – in lines 50 and 51 and maybe elsewhere in the manuscript.

8. Figure 1B in line 154 and 157 is actually Figure 1C.

9. Table 1. Table S1, Figure S1 – It is unclear what N means. It should be clarified how many women with or without XCI-skew were analysed.

10. Please provide in the legend, the numbers of individuals analysed in the three categories in Figure 2A.

11. Please clarify the methodology performed in the sentence stated on lines 181-183: “XCI-skew was associated with increased ASCVD risk score after controlling for BMI and monocyte abundance”.

12. In Figure 3, the cumulative number of events table is misleading (e.g. at 10 years, we have 28 events in the random XCI group and 30 events in the XCI-skew group, and therefore they do not look different, while they are in the graph presenting percentages). It should be indicated the total population analysed per group. Also whether the events occur in distinct individuals or in one individual more than one time should be stated.

13. One aspect that would be important to analyse in the future is the relationship between XCI skewing and mutation burden in blood cells. For example, XCI skew could be a bystander of genetic mutation(s) that would cause clonal expansion of that population. Or XCI skew could determine the fixation of an X-linked mutation in the cell populations. These aspects could be addressed in the discussion.

14. In the general introduction, dosage compensation by XCI and XCI-skewing is not explained in sufficient detail. Several important features, such as the difference between primary and secondary XCI skewing are completely omitted, as well as a clear overview of the clinical relevance of skewed XCI. The main technology employed to assay XCI skewing is also not described. As a consequence, a general readership is likely to not fully appreciate the open questions addressed by authors, and their relevance in light of the current knowledge and available technologies.

15. The authors claim to characterize the relationship of skewed XCI with “molecular, cellular and organismal measures of ageing and cardiovascular disease risk and cancer diagnosis”. However, the correlation between XCI skewing in blood-derived DNA and increased age has been already extensively described (i.e. Bolduc et al., 2008, Busque et al., 1996, Hatakeyama et al., 2004; Zito et al., 2019, and many more). Importantly, with the exception of the assessment of XCI skewing which was obtained for >1.500 individuals, the molecular/cellular data presented by the authors is rather limited. Furthermore, when describing the association between skewed XCI and increased monocyte abundance, the authors omit to discuss that within the blood, different populations of hematopoietic cells show different degrees of XCI skewing with myeloid cells showing a high level of skewed XCI (Gale et al., 1997). The association described by the authors is therefore probably reinforcing these observations, and can likely be explained by biases in cell representation within samples. Furthermore, given the limited number of individuals tested for serum levels of CRP and cytokine interleukin, and the negative association observed for IL-10, these results are difficult to interpret.

16. The relationship between skewed XCI and future cancer diagnosis is also limited by the low numbers of diagnoses in 10-year follow-up and does not allow to take into account cancer type and tissue-specificity. In this context, the authors do not discuss (nor speculate on) how skewed XCI as measured in the blood can be a clinical marker of risk for diseases occurring in other organs. This is a very important point to consider for data interpretation, in light of the existing research addressing the degree of XCI skewing in different tissues and cell types, and not only in blood.

17. To improve the quality of their manuscript to possibly target a broader audience, the authors should more carefully discuss their findings while including relevant references that are omitted in the current version.

18. Lines 7-9.

Age acquired XCI-skew describes the preferential inactivation of one X chromosome across a tissue, which is particularly prevalent in blood tissues of ageing females, and yet its clinical consequences are unknown.

This sentence is misleading. The preferential inactivation of one of the two X chromosomes only occurs at the onset of XCI and results in primary skewed XCI, which influences the choice of which allele becomes inactivated. Age-acquired XCI skewing is a result of post-XCI selection. The authors should include a clear description of the differences between primary and secondary XCI skewing in the introduction, and they should also describe their relevance to human diseases.

19. Line 48-49

X chromosome inactivation (XCI) evolved in placental mammals to compensate for the differences in size between the X and Y sex chromosomes.

The authors should rephrase this sentence. XCI has evolved to compensate for the X-linked gene dosage between XX females and XY males.

20. Lines 85-89

We measured XCI in blood-derived DNA from 1,575 participants (median age = 61) unselected for chronic disease status from the TwinsUK population cohort(Verdi et al., 2019), which comprised of 423 MZ pairs, 257 DZ pairs, and 215 singletons. Using the normalised distribution of XCI, we derive a categorical variable in which XCI-skew equated to 75% XCI, and extreme XCI-skew equated to 91% XCI.

The authors should describe how the degree of XCI skewing was assayed in their experimental settings as a more general readership is not necessarily familiar with the HUMARA assay. A representative image should be shown in a figure, ideally, a main figure but the Supplementary would also work. A detailed description of the method is also required to allow the reader to follow the considerations related to data analysis in the discussion. Indeed the authors are generally far too vague about their approach, even in the Methods section.

21. Line 336-337

The assay failed in 194 samples, and 601 samples were homozygous for the CAG repeat and were therefore uninformative.

The authors should perhaps comment/speculate on why the assay fail on 194/2382 (almost 10%) of samples. Is this a sample-intrinsic issue? e.g. too-low quality/concentration. If not, 10% failure rate is pretty poor.

22. Figures quality and data description in legends should be significantly improved. For example, the authors very often talk about "association" in the manuscript with reference to a suboptimal data representation (e.g. a table) – or without showing the data at all. In my opinion, every such statement should be supported be at least a basic scatter plot to illustrate the trends the authors are talking about (and especially the spread of the data). Especially considering that their figures are extremely small in their current version. On a related note, the authors should move the data from the Supplementary to the main figures. They show so little data in the main figures that they can afford to do it.

23. In figure 2B the labels "lower XCI" and "higher XCI" are misleading.

24. In figure 3, the authors should plot "probability of cancer diagnosis" as referred to in line 209 of the text and provide a comprehensive figure legend.

25. When they described the association between skewed XCI and future cancer diagnosis the authors should break the data down into constituent cancer/organ types. It would be interesting to specifically look into data related to blood cancer – especially considering that they are in many cases sex-biased and also more relatable to their blood-derived XCI-skew measurements. Furthermore, the authors should referred to the well-established links between inflammation and many kinds of cancer if this is the reason why it is meaningful to make more general inferences about cancer from blood measurements.

*Reviewer #1 (Recommendations for the authors):*

Amy L. Roberts et al., investigated the health outcomes impacts of age acquired skewing of X-chromosome inactivation (XCI) ratios occurring in blood cells in 1575 females comprised of 423 monozygotic twins pairs, 257 dyzygotic twins pairs and 215 singleton. They demonstrate: (i) associating between skewing and age; (ii) skewing was independent of other biological markers of aging such as smoking, telomere length, or DNAm Grim Age acceleration or frailty index (iii). (iv) that skewing was associated with a myeloid bias with an increase monocyte to lymphocyte and neutrophil to lymphocyte ratio; (v)a strong negative association with il-10 level and skewing; (vi) skewing was associated with increase cardiovascular risk; and that (vi) skewing was predictive of cancer.

General comments.

Age-associated XCI skewing has been described 25 years ago, yet this phenomenon remains enigmatic in terms of etiology and consequences. The study demonstrates the age-effect on skewing (which is known), but the evolution over time in a sub cohort of subject that were re-sampled. The originality of this study resides in the demonstration that age-associated skewing is associated with health outcomes. The particular strength of the study is the twin component (always aged-matched) and the iterative re-sampling of subgroup of the population study. Despite starting with a large cohort of 1552 individuals, most of the critical observations are made on smaller subsets of subjects: for ASCV risk score the total of subject is 231; IL-10 n=27. Fortunately, for the ASCV score this observation is re-enforced by the twin pairs comparison (N=34). The association with cancer risk is performed on a larger cohort (1417) an controlled for age. However, despite increased risk of Cardiovascular event and cancer, there is no significant association with overall mortality. This indicate that the effect of age-associated skewing is present but modest. The observation that there is a myeloid bias, which is associated with aging, that is more pronounced in subjects with skewing is interesting.

One of the major unknown of this study is how much of the effect attributed to XCI skewing is in fact related to true clonal hematopoiesis or CHIP that is associated with both outcomes described in this study and will mascarade by changes in XCI.

*Reviewer #2 (Recommendations for the authors):*

The manuscript by Roberts et al., is an important piece of work regarding the potentiality of XCI skewing as a biomarker of increase risk for age-driven chronic diseases and cancer. However, the data presenting is exclusively correlative using cohorts of patients.

– The relationship between XCI skewing and ageing in sound (Figure 1A-B). This is the most robust data from this study that confirms previous results (Busque et al., 1996; Gale et al., 1997; Tonno et al., 1998; van Dijk et al., 2002; Zito et al., 2019). It is unquestionable the importance of this result, but as pointed out by the authors, it is not a novel finding.

– The link between XCI skewing and atherosclerotic cardiovascular disease risk or cancer incidence is an interesting one. However, the data is not totally convincing in my view. First, the association with cardiovascular disease (CVD) is based on a risk score and not in actual CVD events. On the other hand, the prospective study of XCI-skew and future cancer diagnosis is based on 58 cancer events. As the authors pointed out, the link between XCI skewing and cancer has been inconsistent in the literature, perhaps because all these studies, including this one, do not reach enough cancer events to draw definitive conclusions. While the prospect of XCI skewing as a biomarker of chronic disease risk and cancer incidence is important, larger cohorts might be needed to ascertain the usefulness of XCI skew as a prognosis marker.

– The data presented is exclusively correlative. Although out of scope of this manuscript, functional experiments addressing the effect of XCI skewing in disease outcome would give insights about a potential causal role of XCI skewing in disease onset/progression/prognosis. These types of experiments will increase the impact of the findings using cohort of patients.

---

## [Author Response]

Essential revisions:1. Was XCI measured on whole blood, versus fraction (PMN, Mononuclear cells, buffy coat, other)?

The DNA used in this study was extracted from whole blood. This has been added to the abstract, method, and results at lines 12-13, 100, and 239.

2. Did the authors look at the correlation between skewing in MZG vs DZ twins. This is known but would be interesting to document.

We have now added the concordance rates to the Results section of the manuscript along with the relevant references:

“In line with previous studies (Vickers et al., 2001), we increased concordance of XCI-skew within MZ twin pairs compared with DZ twin pairs: 27% of MZ twin pairs (114/423), and 45.5% of DZ twin pairs (117/257), were discordant for their categorical XCI-skew status.”

3. Was there enough discordant MZG twin pairs (best controls) for the ASCV risk comparison?

Of the 34 pairs of discordant twins, 17 were DZ and 17 were MZ. The direction of effect is the same across both sets, though of course the power is lower. Using MZs only, we see P = 0.095; and using DZs only, we see P = 0.08.

4. Age is the major confounding factor in all analyses. Although the information is present in the text and methods, I think controlling for age (or age-matched) should also be indicated in all figures.

We have updated all figure legends throughout the manuscript to be clearer. Please see Lines 275-291, 305-310, 340-355, 418-424, 447-454 for all relevant legends, but here we give an example from part of the legend for Figure 3:

“Box plots representing the results of the linear regression mixed effects models to assess (A) Smoking status (P=0.33, N_never_smoker_ = 879; N_ever_smoker_ = 673) and (B) obesity (P=0.88, N_not_obese_ = 726; N_obese_ = 165) after correcting for age with XCI as the dependent variable, and (C) frailty index (P=0.59, N_ranodm XCI_ = 398; n_Skewed XCI_ = 177; N_extreme skew_ = 36), after correcting for age and BMI, (D) Leukocyte telomere length shortening (P=0.9, N_ranodm XCI_ = 278; n_Skewed XCI_ = 103; N_extreme skew_ = 16) after correcting for age and smoking status, and (E) DNAm GrimAge acceleration (P=0.22, N_ranodm XCI_ = 101; n_Skewed XCI_ = 30; N_extreme skew_ = 6), with XCI-skew as the dependent variable.”

5. The hypothesis that cardiovascular and cancer risk could be driven in part by CHIP should be added to the discussion.

We have added a new paragraph to the discussion which addresses this point and the overlap between CHIP and XCI-skew more generally:

“Given the existing links between CHIP and XCI-skew as two age acquired blood traits, and that CHIP mutations are found in individuals with XCI-skew, could the association between XCI-skew and CVD risk be partly driven by CHIP? It is expected that some of the individuals with XCI-skew in our study will also harbour CHIP mutations. However, a better understanding of the co-occurrence of XCI-skew and CHIP within individuals, and the mutational burden within individuals with and without XCI-skew, is an important area of future work. Whether the co-occurrence of XCI-skew and CHIP represents an amplified risk of disease, or helps better define risk categories, will be of clinical significance to establish. As the inherited genetic risk of CHIP is fast becoming better understood, deciphering the genetic predisposition to XCI-skew will also enable the assessment of potential shared genetic susceptibility between the two traits (Kar et al., 2022). However, many of the negative results in our study are crucial findings given they expose differences between the risk factors of XCI-skew and CHIP. Notably, recent work has demonstrated smoking and telomere length are causal risk factors for CHIP (Kar et al., 2022), whereas we see no association with either trait in this study, suggesting CHIP and XCI-skew have some distinct aetiologies.”

6. The XCI skewing data is gathered using the HUMARA test. This is a classical test to score skewing in human cells. Since it is the central method of this manuscript, illustrative examples of the data (random, skew, and extreme skew, as well as non-informative samples) should be provided. This will be helpful to the reader and of broad interest to the eLife audience, especially for (epi)geneticists.

We have added a new Figure 1 to the manuscript with illustrative examples of the raw data from the HUMARA assay representing a random, skewed, and extreme skew sample. We agree this will make the manuscript easier to interpret for a broader readership.

7. When mentioning karyotype, the most common nomenclature is 46, XX, and 46, XY instead of XX,46 or XY,46 – in lines 50 and 51 and maybe elsewhere in the manuscript.

Thank you, the abstract and introduction have now been updated with the correct nomenclature.

8. Figure 1B in lin 154 and 157 is actually Figure 1C.

Thank you for spotting this error. This has been updated. Given the new figures included, this is now Figure 3F.

9. Table 1. Table S1, Figure S1 – It is unclear what N means. It should be clarified how many women with or without XCI-skew were analysed.

We have reorganised the manuscript such that Table 1 and Table S1 are now replaced with more informative figures, as per point 22 below. In each figure legend, as well as the accompanying text, we now specified the sample size with regards to each XCI-skew subgroup. We have left the Supplementary flowchart unchanged for ease of interpretation of overall numbers across the study as we believe the data are now readily available through the manuscript.

10. Please provide in the legend, the numbers of individuals analysed in the three categories in Figure 2A.

The sample sizes have been added to this specific figure legend, as well as to all other legends and Results sections throughout the manuscript.

11. Please clarify the methodology performed in the sentence stated on lines 181-183: "XCI-skew was associated with increased ASCVD risk score after controlling for BMI and monocyte abundance".

We have included the methodology in the ASCVD Results section:

“In a cross-sectional study of 228 individuals (N_ranodm XCI_ = 155; n_Skewed XCI_ = 56; N_extreme skew_ = 17; median age = 62) with matched health data available, XCI-skew was associated with increased ASCVD risk score after controlling for BMI and monocyte abundance using a linear regression mixed effects model (P=0.01, Figure 4A).”

12. In Figure 3, the cumulative number of events table is misleading (e.g. at 10 years, we have 28 events in the random XCI group and 30 events in the XCI-skew group, and therefore they do not look different, while they are in the graph presenting percentages). It should be indicated the total population analysed per group. Also whether the events occur in distinct individuals or in one individual more than one time should be stated.

To carry out this analysis, we excluded any individual with a cancer diagnosis before or within 6 months of the XCI-skew measure to ensure they were cancer free at baseline. We then took the first cancer diagnosis only for each person within up to 10 years of the XCI measure. Therefore, each event in the Cox regression analysis is an independent cancer diagnosis in one person. These details were included in the methods, but we have added them to the figure legend and the results for clarity. The results now read:

“We conducted a prospective 10-year follow-up study (median follow-up time 5.65 years) from time of DNA sampling in 1,417 individuals (N_random_ = 948, N_skewed_ = 469, median age = 60) who were cancer-free at baseline to assess the association between XCI-skew and future cancer diagnoses (cancer events = 58; Supplementary Table S1). Each cancer event represents the first cancer diagnosis (excluding non-melanoma skin cancer) in each individual and subsequent diagnoses are not included.”

We have also added the numbers in each group as well as the percentages of each group with a cancer diagnosis, to the figure legend:

“A Kaplan-Meier plot (top) and cumulative events (bottom) of cancer diagnosis in individuals with XCI-skew (N=469) and random XCI (N=948) are shown. 2.9% (28/948) of individuals with random XCI, and 6.4% (30/469) of individuals with XCI-skew, developed cancer in the 10-year follow-up.”

We have also updated Table S1 to represent the number of cancer diagnoses in each organ/tissue for the Random XCI and XCI-skew group which we hope adds clarity to the data.

13. One aspect that would be important to analyse in the future is the relationship between XCI skewing and mutation burden in blood cells. For example, XCI skew could be a bystander of genetic mutation(s) that would cause clonal expansion of that population. Or XCI skew could determine the fixation of an X-linked mutation in the cell populations. These aspects could be addressed in the discussion.

We agree that this is an important piece of future work. We have added the following paragraph to the discussion which addresses the need for this work:

“Given the existing links between CHIP and XCI-skew as two age acquired blood traits, and that CHIP mutations can be found in individuals with XCI-skew (Busque et al.,. 2012), could the association between XCI-skew and CVD risk be partly driven by CHIP? It is expected that some of the individuals with XCI-skew in our study will also harbour CHIP mutations. However, a better understanding of the co-occurrence of XCI-skew and CHIP within individuals, and the mutational burden within individuals with and without XCI-skew, is an important area of future work. Whether the co-occurrence of XCI-skew and CHIP represents an amplified risk of disease, or helps better define risk categories, will be of clinical significance to establish. As the inherited genetic risk of CHIP is fast becoming better understood, deciphering the genetic predisposition to XCI-skew will also enable the assessment of potential shared genetic susceptibility between the two traits (Kar et al., 2022). However, many of the negative results in our study are crucial findings given they expose differences between the risk factors of XCI-skew and CHIP. Notably, recent work has demonstrated smoking and telomere length are causal risk factors for CHIP (Kar et al., 2022), whereas we see no association with either trait in this study, suggesting CHIP and XCI-skew have some distinct aetiologies.”

14. In the general introduction, dosage compensation by XCI and XCI-skewing is not explained in sufficient detail. Several important features, such as the difference between primary and secondary XCI skewing are completely omitted, as well as a clear overview of the clinical relevance of skewed XCI. The main technology employed to assay XCI skewing is also not described. As a consequence, a general readership is likely to not fully appreciate the open questions addressed by authors, and their relevance in light of the current knowledge and available technologies.

We have expanded the introduction to include examples of primary XCI-skew before introducing secondary XCI-skew, as well as a broadening the topic more generally:

“However, some individuals display a skewed pattern of XCI (XCI-skew), which is defined as a deviation from the expected 1:1 ratio. Examples of primary XCI-skew have been identified, including stochastic events resulting in XCI-skew across all tissues (Tukiainen et al., 2016) or the preferential selection of cells expressing functioning alleles in heterozygous females with X-linked recessive traits (Busque and Gilliland, 1998; Nyhan et al., 1970). However, secondary or age acquired XCI-skew is more common and refers to increasing XCI-skew with age, particularly in mitotically active blood tissue (Busque et al., 1996; Gale et al., 1997).

We have also provided a new Figure 1 which details how XCI-skew is derived from the HUMARA assay (as per point 6 above), and included the following statement in the first Results section regarding the correlation of HUMARA with transcription-based methods:

“We measured XCI in DNA derived from whole blood using the methylation-sensitive PCR-based Human Androgen Receptor Assay (HUMARA) (Cutler Allen et al., 1992; Hatakeyama et al., 2004) which differentiates between alleles from the active and inactive X (Cutler Allen et al., 1992; Hatakeyama et al., 2004). HUMARA is an extensively used assay which correlates well with transcription-based methods (Bolduc et al., 2008; Zito et al., 2019).*”*

15. The authors claim to characterize the relationship of skewed XCI with "molecular, cellular and organismal measures of ageing and cardiovascular disease risk and cancer diagnosis". However, the correlation between XCI skewing in blood-derived DNA and increased age has been already extensively described (i.e. Bolduc et al., 2008, Busque et al., 1996, Hatakeyama et al., 2004; Zito et al., 2019, and many more). Importantly, with the exception of the assessment of XCI skewing which was obtained for >1.500 individuals, the molecular/cellular data presented by the authors is rather limited. Furthermore, when describing the association between skewed XCI and increased monocyte abundance, the authors omit to discuss that within the blood, different populations of hematopoietic cells show different degrees of XCI skewing with myeloid cells showing a high level of skewed XCI (Gale et al., 1997). The association described by the authors is therefore probably reinforcing these observations, and can likely be explained by biases in cell representation within samples. Furthermore, given the limited number of individuals tested for serum levels of CRP and cytokine interleukin, and the negative association observed for IL-10, these results are difficult to interpret.

We hope it was clear in our manuscript that the association seen with ageing is a replication of the many studies that predate ours, and not a novel finding. See lines 463-465 of discussion, which are unchanged from the original version, but which introduce the topic in reference to the long-established association with increasing age. We feel that no study on XCI-skew would be complete without a summary of the association with age seen in this cohort. However, we have amended the manuscript throughout to ensure its as clear as possible that the age association is an expected result, and acts as a validation of our dataset in line with the existing literature. The results now read as follows:

“As expected, after controlling for relatedness and zygosity, we find a significant positive association between age and XCI skewing (P=2.8x10^-9^, N=1,575). This result replicates the many existing studies on age acquired XCI-skew and acts as a validation of the TwinsUK HUMARA dataset (Busque et al., 1996; Gale et al., 1997; Hatakeyama et al., 2004; Zito et al., 2019).”

We have also amended the statement in our introduction so that it now reads:

“We tested this hypothesis by assaying XCI-skew in 1,575 females from the TwinsUK cohort and employed prospective, cross-sectional, and intra-twin designs, to characterise the relationship of XCI with molecular and cellular measures of ageing, cardiovascular disease risk, and cancer diagnoses.”

We have amended the discussion include reference to the Gale 1997 study, in which there is lower levels of skew in T lymphocytes compared to monocytes, thought to be due to the longer life span of these lymphoid cells. This is appended to our existing discussion point in which we reference two other studies which show correlation between the skew of different cell types.

“Furthermore, age acquired XCI-skew has previously been shown across isolated neutrophils, monocytes, and T cells, with correlations between these fractions(Tonon et al., 1998; van Dijk et al., 2002), albeit with lower levels of skewing in lymphoid cells (Gale et al., 1997).”

We agree that the cytokine data are limited, and though we had addressed this in the discussion (see Lines 566-568), we have now updated the results to better reflect this too:

“We date-matched the XCI data with serum levels of CRP (N_ranodm XCI_ = 121; n_Skewed XCI_ = 38; N_extreme skew_ = 6) and a more modest cytokine dataset of interleukin (IL)-6, IL-1B, IL-10, and TNF (N_ranodm XCI_ = 23; n_Skewed XCI_ = 4) and used linear regression mixed effects models to control for batch effects, age, seasonality, family structure and zygosity (Supplementary Figure S3). We see no association with primary markers of inflammageing CRP (P=0.41), IL-6 (P=0.41), or TNF (P=0.61), though a nominal association with IL-1β is observed (P=0.02) which does not pass multiple correction, but warrants follow up analysis with a larger sample size.”

16. The relationship between skewed XCI and future cancer diagnosis is also limited by the low numbers of diagnoses in 10-year follow-up and does not allow to take into account cancer type and tissue-specificity. In this context, the authors do not discuss (nor speculate on) how skewed XCI as measured in the blood can be a clinical marker of risk for diseases occurring in other organs. This is a very important point to consider for data interpretation, in light of the existing research addressing the degree of XCI skewing in different tissues and cell types, and not only in blood.

With 28 cancer events in 948 individuals with random XCI, and 30 cancer events in 469 individuals with XCI-skew, we believe this is a well-powered study for total cancer risk. However, we do agree that we were not sufficiently powered to take into account the tissue-specificity, and it would be interesting to explore cancer risk in specific tissues. Of note, see point 25 below in which removing Haematopoietic / Lymphoid Tissues cancers slightly strengthens the association observed.

Notwithstanding the above, we have added a new paragraph to the discussion in which we speculate on these associations and how they may likely be driven by inflammation and are unlikely to be causally driven by XCI-skew:

“Though CHIP is also predictive of future cancer diagnosis, the association is limited to haematological cancers (Desai et al., 2018). Whereas here we see blood-derived measures of XCI-skew is predictive of all future cancer diagnoses, even when the haematological cancers are removed from the analysis. Follow-up studies to assess the risk of cancer of specific tissues in individuals with XCI-skew are needed, but we hypothesise that the relationship between XCI-skew measures in blood tissue and cancer will not be causal. Instead XCI-skew is likely a marker of chronic inflammation, which can predispose to the development of cancer through increased mutagenesis and can promote tumorigenesis by shaping the tumour microenvironment to stimulate tumour growth (Greten and Grivennikov, 2019). However, it is interesting that common environmental factors that induce chronic inflammation, such as smoking and obesity, have not been found to be risk factors for XCI-skew in our study.”

17. To improve the quality of their manuscript to possibly target a broader audience, the authors should more carefully discuss their findings while including relevant references that are omitted in the current version.

We have extended the discussion (see points 13 and 16 above) and included additional references throughout. We hope the reviewer agrees this makes the manuscript more relevant to a broader audience.

18. Lines 7-9.Age acquired XCI-skew describes the preferential inactivation of one X chromosome across a tissue, which is particularly prevalent in blood tissues of ageing females, and yet its clinical consequences are unknown.This sentence is misleading. The preferential inactivation of one of the two X chromosomes only occurs at the onset of XCI and results in primary skewed XCI, which influences the choice of which allele becomes inactivated. Age-acquired XCI skewing is a result of post-XCI selection. The authors should include a clear description of the differences between primary and secondary XCI skewing in the introduction, and they should also describe their relevance to human diseases.

We have amended this sentence in the abstract:

“Age acquired XCI-skew describes the preferential selection of cells across a tissue resulting in an imbalance of XCI, which is particularly prevalent in blood tissues of ageing females and yet its clinical consequences are unknown.”

We have also expanded the introduction to include primary XCI skew too (see also point 14 above).

“However, some individuals display a skewed pattern of XCI (XCI-skew), which is defined as a deviation from the expected 1:1 ratio. Examples of primary XCI-skew have been identified, including stochastic events resulting in XCI-skew across all tissues (Tukiainen et al., 2016) or the preferential selection of cells expressing functioning alleles in heterozygous females with X-linked recessive traits (Busque and Gilliland, 1998; Nyhan et al., 1970). However, secondary or age acquired XCI-skew is more common and refers to increasing XCI-skew with age, particularly in mitotically active blood tissue (Busque et al., 1996; Gale et al., 1997). Within individuals, the correlation of XCI ratios between blood and other tissues diminishes over the life course as the XCI ratios in blood continue to skew with age (Bolduc et al., 2008; Zito et al., 2019).”

19. Line 48-49X chromosome inactivation (XCI) evolved in placental mammals to compensate for the differences in size between the X and Y sex chromosomes.The authors should rephrase this sentence. XCI has evolved to compensate for the X-linked gene dosage between XX females and XY males.

We agree and have changed the sentence on lines 52-53 as suggested.

20. Lines 85-89We measured XCI in blood-derived DNA from 1,575 participants (median age = 61) unselected for chronic disease status from the TwinsUK population cohort(Verdi et al., 2019), which comprised of 423 MZ pairs, 257 DZ pairs, and 215 singletons. Using the normalised distribution of XCI, we derive a categorical variable in which XCI-skew equated to 75% XCI, and extreme XCI-skew equated to 91% XCI.The authors should describe how the degree of XCI skewing was assayed in their experimental settings as a more general readership is not necessarily familiar with the HUMARA assay. A representative image should be shown in a figure, ideally, a main figure but the Supplementary would also work. A detailed description of the method is also required to allow the reader to follow the considerations related to data analysis in the discussion. Indeed the authors are generally far too vague about their approach, even in the Methods section.

We have now included a new Figure 1 in the manuscript which represents how the raw HUMARA assay data is used to calculate XCI skew (see also point 6 above). We have also included an additional Supplementary Figure S1 which demonstrates how the categorical variables were derived from the continuous data, as well as added the following text to the Results section for clarity:

“The output of the HUMARA assay is a continuous XCI variable from 0-100%, where 50% is perfectly balanced XCI and the directionality of XCI away from 50% is uninformative (e.g., both 0% and 100% are considered equal). We normalised the distribution of the continuous XCI values across the cohort, and defined XCI-skew as measures 1s.d. from the mean, corresponding to XCI score ≥75%, and extreme XCI-skew as measures 2s.d. from the mean, corresponding to XCI score ≥91% (Supplementary Figure S1).”

We have also expanded the HUMARA methods section to allow a fuller understanding of the assay:

“The HUMARA method which combines methylation-sensitive restriction enzyme digest and amplification of a highly polymorphic (CAG)n repeat in the first exon of the X-linked AR gene, allowing for the differentiation of the active and inactive chromosomes in heterozygous individuals (Cutler Allen et al., 1992). Here, 625ng of genomic DNA was divided into three aliquots and incubated for 30 minutes at 37°C with (i) the methylation-sensitive enzyme HpaII, (ii) the methylation-insensitive enzyme MspI, or (iii) water (mock digest) in 1x New England Biolabs CutSmart Buffer. The HpaII digest was followed by an additional 20 minutes at 80°C to avoid residual enzymatic activity. Fluorescently labelled PCR primers (FAM, VIC, NED, or PET; Forward primer 5’-dye-GCTGTGAAGGTTGCTGTTCCTCAT-3’, Reverse primer 5’-TCCAGAATCTGTTCCAGAGCGTGC-3’) were used in New England BioLabs One Taq Master Mix to amplify 1.5μl of digested PCR product. The Mock and HpaII digested DNA were amplified in triplicate (using FAM, VIC, and NED), and the MspI digest, used as control of digestion efficiency, was amplified once (using PET). All PCRs were amplified with an initial denaturation step at 94°C for 5 mins, followed by 30 cycles of 94°C for 30 secs, 60°C for 1 min, and 72°C for 2 mins, and a final elongation step of 72°C for 15 mins.”

21. Line 336-337The assay failed in 194 samples, and 601 samples were homozygous for the CAG repeat and were therefore uninformative.The authors should perhaps comment/speculate on why the assay fail on 194/2382 (almost 10%) of samples. Is this a sample-intrinsic issue? e.g. too-low quality/concentration. If not, 10% failure rate is pretty poor.

We do indeed suspect most assay failures were due to underlying low DNA concentrations of the archived, biobanked samples used in the study. We did not observe any batch effects with the failure rates that would suggest experimental issues. Furthermore, samples typically failed across all three PCRs (FAM, VIC, and NED) also suggesting intrinsic issues which typically impact PCRs, such as DNA concentration. Though the best effort was taken to standardise the DNA concentration of the samples prior to amplification, we suspect DNA concentrations will be a major cause of sample failures.

22. Figures quality and data description in legends should be significantly improved. For example, the authors very often talk about "association" in the manuscript with reference to a suboptimal data representation (e.g. a table) – or without showing the data at all. In my opinion, every such statement should be supported be at least a basic scatter plot to illustrate the trends the authors are talking about (and especially the spread of the data). Especially considering that their figures are extremely small in their current version. On a related note, the authors should move the data from the Supplementary to the main figures. They show so little data in the main figures that they can afford to do it.

We have included two new main figures (see Figure 1 and Figure 3), and two new supplementary figures (see Supplementary Figure S1 and Supplementary Figure S3) to better present the data, particularly replacing Table 1 and Table S1 with figures (see also point 9 above). We have also re-written all figure legends to ensure the methodology is clear to the reader.

23. In figure 2B the labels "lower XCI" and "higher XCI" are misleading.

The discordant twins can be any combination of the 3 categorical variables: random XCI and skewed XCI, random XCI and extreme skewed XCI, or skewed XCI and extreme skewed. Therefore, we found it clearest to label the x-axis as “lower” and “higher” as the groups are somewhat heterogenous. However, we have amended these labels to read “twin with lower skewed XCI” and “twin with higher skewed XCI”, which we hope adds clarity. We would be very happy to use a different term here if the reviewer has other suggestions.

24. In figure 3, the authors should plot "probability of cancer diagnosis" as referred to in line 209 of the text and provide a comprehensive figure legend.

The axis label for Figure 5 (note, previously figure 3) has been changed, and the figure legend (as per comments 4 and 12) has been improved.

25. When they described the association between skewed XCI and future cancer diagnosis the authors should break the data down into constituent cancer/organ types. It would be interesting to specifically look into data related to blood cancer – especially considering that they are in many cases sex-biased and also more relatable to their blood-derived XCI-skew measurements. Furthermore, the authors should referred to the well-established links between inflammation and many kinds of cancer if this is the reason why it is meaningful to make more general inferences about cancer from blood measurements.

The origin of the cancers is included in “Supplementary Table S1: Cancer diagnoses recorded in 10-year follow-up by organ/site.”. We have extended this table to show the cancer diagnoses per XCI random and XCI skewed group too. We are underpowered to run the association on subsets of cancers, however, re-running the analysis after first excluding the Haematopoietic / Lymphoid Tissues cancers slightly strengthens the association observed (P=0.009; HR=2.04 (1.20-3.49)) compared to the original analysis (P=0.012; HR = 1.95 (1.16-3.28)). This suggests the association is not driven by the small number of Haematopoietic / Lymphoid Tissues in the dataset. We have added this to the results (Lines 440-443) and the discussion (Lines 496-498).

“Though we were underpowered to run associations for specific cancers, given the robust association of CHIP with haematological cancers we were interested to explore the potential that blood cancers were partly driving the association. However, running the analysis without the Haematopoietic/Lymphoid Tissue cancers (see Supplementary Table S1) strengthened the observed association (P=0.009; HR = 2.04 (1.20-3.49)) suggesting a crucial difference in cancer risk between XCI-skew and CHIP.”

We have also extended the discussion to include the role of inflammation in cancer, and how this may be the link between XCI-skew in blood and cancer across various tissues.

“Though CHIP is also predictive of future cancer diagnosis, the association is limited to haematological cancers (Desai et al., 2018). Whereas here we see blood-derived measures of XCI-skew is predictive of all future cancer diagnoses, even when the haematological cancers are removed from the analysis. Follow-up studies to assess the risk of cancer of specific tissues in individuals with XCI-skew are needed, but we hypothesise that the relationship between XCI-skew measures in blood tissue and cancer will not be causal. Instead XCI-skew is likely a marker of chronic inflammation, which can predispose to the development of cancer through increased mutagenesis and can promote tumorigenesis by shaping the tumour microenvironment to stimulate tumour growth (Greten and Grivennikov, 2019). However, it is interesting that common environmental factors that induce chronic inflammation, such as smoking and obesity, have not been found to be risk factors for XCI-skew in our study.”

Reviewer #1 (Recommendations for the authors):Amy L. Roberts et al., investigated the health outcomes impacts of age acquired skewing of X-chromosome inactivation (XCI) ratios occurring in blood cells in 1575 females comprised of 423 monozygotic twins pairs, 257 dyzygotic twins pairs and 215 singleton. They demonstrate: (i) associating between skewing and age; (ii) skewing was independent of other biological markers of aging such as smoking, telomere length, or DNAm Grim Age acceleration or frailty index (iii). (iv) that skewing was associated with a myeloid bias with an increase monocyte to lymphocyte and neutrophil to lymphocyte ratio; (v)a strong negative association with il-10 level and skewing; (vi) skewing was associated with increase cardiovascular risk; and that (vi) skewing was predictive of cancer.General comments.Age-associated XCI skewing has been described 25 years ago, yet this phenomenon remains enigmatic in terms of etiology and consequences. The study demonstrates the age-effect on skewing (which is known), but the evolution over time in a sub cohort of subject that were re-sampled. The originality of this study resides in the demonstration that age-associated skewing is associated with health outcomes. The particular strength of the study is the twin component (always aged-matched) and the iterative re-sampling of subgroup of the population study. Despite starting with a large cohort of 1552 individuals, most of the critical observations are made on smaller subsets of subjects: for ASCV risk score the total of subject is 231; IL-10 n=27. Fortunately, for the ASCV score this observation is re-enforced by the twin pairs comparison (N=34). The association with cancer risk is performed on a larger cohort (1417) an controlled for age. However, despite increased risk of Cardiovascular event and cancer, there is no significant association with overall mortality. This indicate that the effect of age-associated skewing is present but modest. The observation that there is a myeloid bias, which is associated with aging, that is more pronounced in subjects with skewing is interesting.One of the major unknown of this study is how much of the effect attributed to XCI skewing is in fact related to true clonal hematopoiesis or CHIP that is associated with both outcomes described in this study and will mascarade by changes in XCI.

Thank you for your helpful comments. Though we agree that sample sizes are low in some analyses (which we now address more carefully in the revised manuscript), we would also like to draw attention to the many analyses with robust sample sizes, such as smoking (n=1,552), telomere length (n=397), frailty (n=611) and obesity (n=891). These negative findings are vital in our understanding of the causes and consequences of XCI-skew, suggesting XCI-skew as an independent biomarker.

We completely agree that it is of upmost importance to establish the link with CHIP more robustly, and we hope to do this in follow-up work. However, based on existing literature and our current study, there are some key differences between XCI-skew and CHIP. Firstly, telomere length and smoking are both been identified as causal risk factors for CHIP (Kar, 2022), yet we see no such association in our work. And secondly, CHIP is exclusively associated with blood cancers, whereas we see an association with cancer across all tissues/organs even when removing the small number of haematopoietic cancers from the dataset (n=4 cases). Together these observations suggest that robustly defining the relationship between XCI-skew and CHIP is of vital importance as we do not believe one is fully explained by the other. We have updated the manuscript to draw attention to these important differences. What will be of great interest is determining if the presence of both CHIP and XCI-skew within an individual poses a greater disease risk than either molecular trait occurring independently. We have added these points to the discussion of the paper (see Lines 522-537) which we believe improves the manuscript and we thank you once again for your comments.

Reviewer #2 (Recommendations for the authors):The manuscript by Roberts et al., is an important piece of work regarding the potentiality of XCI skewing as a biomarker of increase risk for age-driven chronic diseases and cancer. However, the data presenting is exclusively correlative using cohorts of patients.– The relationship between XCI skewing and ageing in sound (Figure 1A-B). This is the most robust data from this study that confirms previous results (Busque et al., 1996; Gale et al., 1997; Tonno et al., 1998; van Dijk et al., 2002; Zito et al., 2019). It is unquestionable the importance of this result, but as pointed out by the authors, it is not a novel finding.– The link between XCI skewing and atherosclerotic cardiovascular disease risk or cancer incidence is an interesting one. However, the data is not totally convincing in my view. First, the association with cardiovascular disease (CVD) is based on a risk score and not in actual CVD events. On the other hand, the prospective study of XCI-skew and future cancer diagnosis is based on 58 cancer events only. As the authors pointed out, the link between XCI skewing and cancer has been inconsistent in the literature, perhaps because all these studies, including this one, do not reach enough cancer events to draw definitive conclusions. While the prospect of XCI skewing as a biomarker of chronic disease risk and cancer incidence is important, larger cohorts might be needed to ascertain the usefulness of XCI skew as a prognosis marker.– The data presented is exclusively correlative. Although out of scope of this manuscript, functional experiments addressing the effect of XCI skewing in disease outcome would give insights about a potential causal role of XCI skewing in disease onset/progression/prognosis. These types of experiments will increase the impact of the findings using cohort of patients.

Thank you for your helpful comments. We agree that assessing the risk of XCI-skew on incidence CVD is of upmost importance to establish the true risk of this age acquired trait and this has now been emphasised further in our discussion.

However, we believe our cancer analysis to be more robust than previous studies for three main reasons: 1) we have corrected for age at study entry in our model; 2) all individuals were cancer-free at baseline therefore removing confounding factors of treatments; 3) we did not focus on cancers of the female reproductive system only as we do not believe this is a robust a priori hypothesis. However, we fully agree with the reviewer that larger studies would ideally be required to confirm the usefulness of XCI-skew as a prognosis marker.

We also agree with the need for functional experiments, and we hope our study demonstrating these correlations will enable this crucial follow-up work to take place. We would also like to emphasise once more that many of the robust analyses in our study with negative results are of great importance when considering follow-up work. For instance, we have shown that XCI-skew does not correlate with other cellular and molecular markers of biological ageing, nor indeed that frailty, smoking nor obesity are risk factors, and these results we feel are highly significant findings.